# SmokeViz: A Large-Scale Satellite Dataset for Wildfire Smoke Detection and Segmentation

**Rey Koki**[1,2,3]    **Michael McCabe**[3,4]    **Josh Myers-Dean**[5]    **Dhruv Kedar**[3]
**Annabel Wade**[1,6]    **Jebb Q. Stewart**[1]    **Christina Kumler-Bonfanti**[1,2,3]    **Jed Brown**[3]

[1]NOAA Global Systems Laboratory    [2]Cooperative Institute for Research in Environmental Sciences
[3]University of Colorado Boulder    [4]Flatiron Institute    [5]Adobe    [6]University of Washington
rey.koki@colorado.edu

## Abstract

The global rise in wildfire frequency and intensity over the past decade underscores the need for improved fire monitoring techniques. To advance deep learning research on wildfire detection and its associated human health impacts, we introduce **SmokeViz**, a large-scale machine learning dataset of smoke plumes in satellite imagery. The dataset is derived from expert annotations created by smoke analysts at the National Oceanic and Atmospheric Administration, which provide coarse temporal and spatial approximations of smoke presence. To enhance annotation precision, we propose **pseudo-label dimension reduction (PLDR)**, a generalizable method that applies pseudo-labeling to refine datasets with mismatching temporal and/or spatial resolutions. Unlike typical pseudo-labeling applications that aim to increase the number of labeled samples, PLDR maintains the original labels but increases the dataset quality by solving for intermediary pseudo-labels (IPLs) that align each annotation to the most representative input data. For SmokeViz, a parent model produces IPLs to identify the single satellite image within each annotations time window that best corresponds with the smoke plume. This refinement process produces a succinct and relevant deep learning dataset consisting of over 160,000 manual annotations. The SmokeViz dataset is expected to be a valuable resource to develop further wildfire-related machine learning models and is publicly available at `https://noaa-gsl-experimental-pds.s3.amazonaws.com/index.html#SmokeViz/`.

## 1 Introduction

Due in part to public policy, average fine particulate matter ($PM_{2.5}$) levels in the United States have declined over recent decades [1]. However, from 2010 to 2020, the contribution of wildfire smoke to $PM_{2.5}$ concentrations more than doubled, accounting for up to half of total $PM_{2.5}$ exposure in Western United States [2]. This is particularly concerning, as ambient $PM_{2.5}$ is a leading environmental risk factor for adverse health outcomes and premature mortality [3]. These trends/risks highlight the urgent need for scalable and timely smoke monitoring systems to mitigate public health risks.

Satellite imagery offers the spatial coverage and temporal frequency needed for large-scale smoke monitoring. In comparison to polar-orbiting satellites like Suomi or Sentinel, geostationary satellites such as the GOES series [4] are especially well-suited to this task, providing persistent observation over fixed regions, essential for capturing the dynamic behavior of wildfire smoke plumes. The high temporal resolution and wide coverage of GOES imagery enable real-time tracking of smoke concentration and movement, supporting air quality assessments and early warning systems.

Even with the advances in remote sensing, existing deep learning satellite datasets for wildfire smoke detection face several limitations. They are often small in scale, restricted to specific regions or events,

and focus on scene-level classification rather than pixel-level segmentation. Most do not differentiate between smoke density levels, are not publicly available, and lack standardized benchmarks for semantic segmentation. While NOAA's Hazard Mapping System (HMS) provides a large-scale, expert-labeled dataset, its annotations span multi-hour time windows that vary in duration. This creates a temporal mismatch between the labels and individual satellite frames, complicating their direct use for supervised learning.

To address these challenges, we introduce **SmokeViz**, a large-scale satellite dataset for semantic segmentation of wildfire smoke plumes. SmokeViz includes over 160,000 annotated samples derived from GOES-East and GOES-West imagery, aligned with HMS analyst annotations. To resolve the temporal ambiguity in the original labels, we propose a semi-supervised method called **pseudo-label dimension reduction (PLDR)**, which uses intermediary pseudo-labels to select the satellite image that best matches each smoke annotation. The resulting dataset provides one-to-one image-to-label pairs with ordinal smoke density masks, suitable for supervised deep learning.

**SmokeViz** serves as a benchmark for wildfire smoke segmentation and as a resource for the broader machine learning community working with geospatial, temporal, and remote sensing data. It supports new directions in ordinal segmentation, semi-supervised learning with temporal uncertainty, and pre-training for Earth observation tasks involving dynamic atmospheric phenomena.

The contributions presented in this paper include **SmokeViz**, the largest satellite-based dataset for wildfire smoke segmentation, with over 160,000 samples from GOES imagery, our proposed **PLDR**, a physics-guided semi-supervised method for aligning coarse human annotations with temporally optimal satellite imagery and benchmark segmentation baselines with standardized training splits to support reproducibility and future studies.

## 2   Related Work

### 2.1   Smoke Detection and Labeling Methods

Multi-channel thresholding remains a widely used method for distinguishing smoke from similar atmospheric signatures such as dust or clouds using channel-specific radiance values [5]. These thresholds are typically derived from labeled historical data and are fine-tuned to specific regions and fuel types, limiting their generalizablity [6]. In contrast, the SmokeViz dataset spans a wide range of biogeographies across North America and can serve as a source of refined analyst-labeled examples for developing more generalizable thresholding techniques.

Large parameterized numerical models are used for forecasting smoke dispersion, but not for smoke detection itself. Systems such as HRRR-Smoke and RRFS [7, 8] rely on computationally intensive forecasts requiring nearly 200 dynamic meteorological inputs. A key limitation of these models is the absence of a real-time smoke analysis product for data assimilation, resulting in delayed model spin-up and compounded forecast errors. Model predictions from SmokeViz could help fill this gap, offering a real-time, satellite-driven alternative to support data assimilation for operational smoke dispersion forecasting.

Manual smoke labeling is performed by trained analysts through visual inspection of satellite imagery. NOAA's Hazard Mapping System (HMS) provides a analyst-labeled wildfire smoke dataset [9, 10]. HMS analysts examine GOES imagery sequences to track smoke plume movement and annotate the approximate spatial extent and qualitative density of smoke (light, medium, heavy), as illustrated in Figure 2.1. Annotations are issued on a rolling basis and span time windows ranging from instantaneous to over 20 hours [11]. While HMS provides high-quality expert annotations, its operational format introduces challenges for supervised learning: annotations are temporally coarse, vary in length, and lack one-to-one correspondence with satellite frames. SmokeViz refines HMS annotations into temporally resolved, frame-aligned labels, enabling real-time, continuous predictions of smoke extent and density.

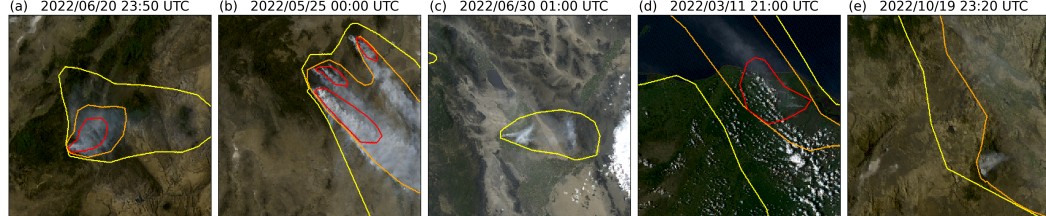

Figure 1: HMS smoke annotations overlaid on GOES imagery. Yellow, orange, and red contours indicate light, medium, and heavy smoke density, respectively. (a) and (b) show canonical smoke plumes; (c)–(e) illustrate density label variation across scenes.

## 2.2 Deep Learning Datasets and Models for Wildfire Smoke

Recent efforts have applied deep learning to wildfire smoke detection using a variety of satellite sources and label strategies. SmokeNet [12] employs a convolutional neural network (CNN) to classify MODIS image scenes as containing smoke or not, using student-provided labels. SatlasPretrain [13] includes a small set of Sentinel-2 images labeled for smoke as part of a larger multi-label pre-training dataset. While scene classification methods can provide wildfire detection information, they do not capture spatial characteristics of smoke plumes that segmentation would be more appropriate to capture.

Several datasets have been developed for smoke segmentation, but they are limited in scope. Wen et al. [14] trained a CNN on GOES-East imagery over California and Nevada using HMS annotations from the 2018 wildfire season. Larsen et al. [15] used Himawari-8 data to detect smoke at the pixel level for a single fire event, using a threshold-based algorithm as ground truth. Table 1 compares these datasets in terms of scale, source, and labeling. SmokeViz stands out by offering over 160,000 samples with analyst-generated, frame-aligned labels covering multiple fire seasons, regions, and biogeographies. Not only do we use geostationary satellites with persistent observations, but we choose either GOES-East or GOES-West based on which satellite has optimal observational conditions of the event. It is, to our knowledge, the largest and most diverse dataset for smoke plume segmentation.

Table 1: Comparison of satellite smoke plume datasets, detailing the number of smoke plume samples, satellite source (polar orbiting (P) or geostationary (G)), number of spectral bands, labeling method, classification type - scene classification (SC) or semantic segmentation (SS), and public availability.

| reference | # samples | satellite | # bands | label | task | avail. |
|---|---|---|---|---|---|---|
| [12] | 1016 | MODIS (P) | 5 | students | SC | no |
| [13] | 125 | Sentinel-2 (P) | 3 | crowd sourced | SC | yes |
| [14] | 4095 | GOES-East (G) | 5 | HMS analysts | SS | no |
| [15] | 975 | Himiwari-8 (G) | 7 | algorithm | SS | no |
| SmokeViz | 163,479 | GOES-East+West (G) | 3 | HMS analysts | SS | yes |

In addition to its relevance for wildfire applications, SmokeViz contributes a challenging benchmark for general-purpose remote sensing vision tasks. Unlike many existing datasets that avoid cloudy scenes [16, 17] or focus on sharply bounded features such as cropland [17], infrastructure [18], or oceanic clouds [19, 20], smoke has amorphous, fading boundaries in both space and time. Incorporating smoke segmentation into large-scale pre-training corpora, such as SatlasPretrain [13], could enhance generalizable models for Earth observation.

## 2.3 Pseudo-labeling and Semi-Supervised Learning

Semi-supervised learning techniques such as pseudo-labeling have been widely used to expand training data by leveraging unlabeled samples [21]. Typically, a parent model is trained on labeled data and then used to generate pseudo-labels for an unlabeled dataset, which are in turn used to train subsequent models in an iterative process.

In contrast, we propose a non-iterative variation focused not on data expansion, but dataset data-to-label precision. Our method, **pseudo-label dimension reduction (PLDR)**, generates intermediary pseudo-labels (IPLs) for each satellite frame within the HMS annotation window. Rather than using these labels for training, we use them to identify the satellite image with the greatest alignment to the analyst annotation. This enables the construction of SmokeViz, a temporally disambiguated, one-to-one image-to-label dataset. The resulting dataset methodically pairs the analyst-generated smoke plume labels with selected GOES imagery, enabling high-resolution, temporally accurate segmentation model training.

Beyond wildfire smoke segmentation, PLDR offers a general framework for aligning coarse or weakly matched datasets. This is particularly useful in domains such as remote sensing, medical imaging, and video analysis, where annotations often span temporal or spatial intervals rather than individual frames. In Earth observation specifically, atmospheric parameters are often combined from disparate sources with inconsistent spatial and temporal resolutions, making it difficult to integrate them into unified training datasets. By using intermediary pseudo-labels to identify the most representative input sample, PLDR transforms many-to-one or one-to-many supervision into clean one-to-one mappings. This enables more precise alignment between data and labels, facilitating integration across heterogeneous sources without requiring additional hand-labeling. As presented, PLDR serves as a practical preprocessing strategy for repurposing historical legacy datasets with temporal ambiguity into precise training resources for modern deep learning models.

## 3   Methods

### 3.1   Datasets

We use imagery from the latest GOES satellites: GOES-16 (East), GOES-17, and GOES-18 (West), each equipped with the Advanced Baseline Imager (ABI), which captures 16 spectral bands from visible to infrared wavelengths every 10 minutes. We process bands 1-3 using PyTroll [22] to generate 1km true-color composites [23], matching the imagery reviewed by HMS analysts. These bands correspond to the shortest wavelengths available on ABI and yield the highest signal-to-noise ratio (SNR).

To approximate the dynamic movement of smoke, HMS analysts annotate plumes using multi-frame satellite animations. These annotations span varying time windows, averaging three hours. Since the HMS annotations are designed to reflect overall plume extent during a time window rather than at any specific moment, smoke boundaries in individual frames may not align well with the annotation (Figure 2). A naive modeling approach would use all frames within each time window as input, but this introduces non-uniform sequence lengths and significantly increases memory and computational demands and complicates the use of CNN architectures. Instead, we establish a one-to-one mapping by identifying the single satellite frame that best matches each analyst annotation.

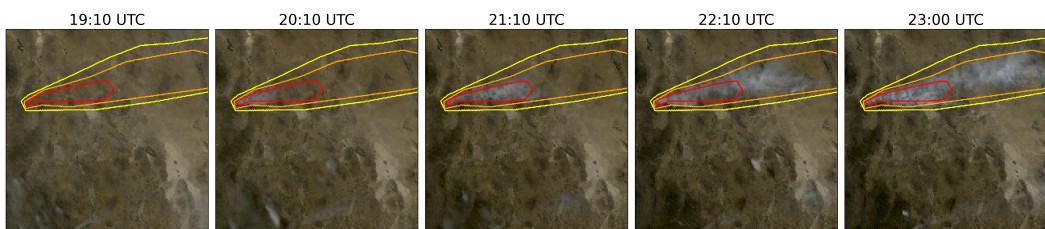

Figure 2: True color GOES-East imagery from May 5th, 2022, Southeast New Mexico (31.38°N, 107.87°W) during the start of the Foster Fire. The red, orange and yellow lines represent the heavy, medium and low density HMS smoke annotations that span 19:10–23:00 UTC.

We select either GOES-East or GOES-West based on the solar zenith angle (SZA) to optimize for forward Mie scattering, which enhances smoke visibility in satellite imagery. Smoke particles (100nm-10μm) scatter light predominantly via Mie scattering when $\lambda < d$, favoring short wavelengths and forward angles (Figure 3). To generate the Mie-derived dataset, we evaluate the available satellite platforms for each annotation time window and choose the satellite (East or West) that is expected

to observe the strongest forward scattering geometry based on sun-satellite alignment. This ensures selection of the satellite view with the highest potential smoke SNR if smoke were present. Therefore, we select (1) the satellite expected to yield the strongest Mie forward scattering (Figures 4(a) vs 4(b)) and (2) the three shortest wavelength ABI bands (C01-C03: 0.47, 0.64, and 0.865μm) (Figures 4(c)-4(e)).

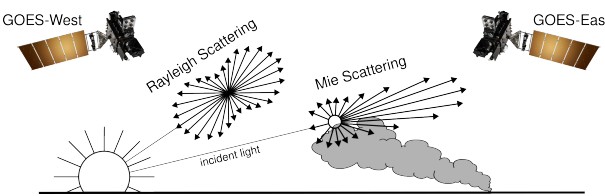

Figure 3: If the particle size is $< \frac{1}{10}$ the $\lambda$ of the interacting light, then the primary scattering will be Rayleigh. Mie scattering is the predominant scattering mechanism when the particle size is larger than the $\lambda$ of light. This schematic demonstrates that when the sun is setting in the West, the Mie scattering will predominately forward scatter towards GOES-East.

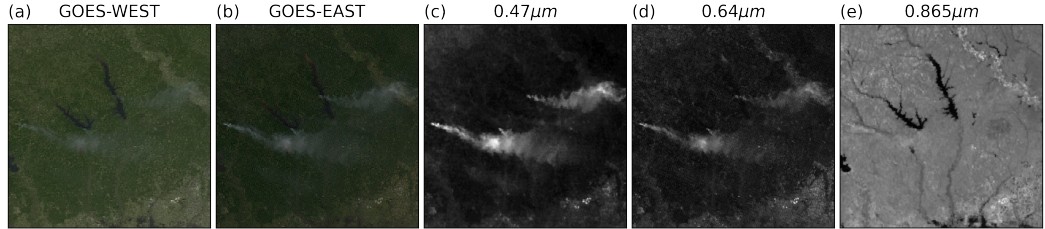

Figure 4: True color (a) GOES-WEST and (b) GOES-EAST imagery from March $23^{rd}$, 2022 centered at $(31.1°, -93.8°)$ in Texas, USA taken at 23:20 UTC. The GOES-EAST raw band imagery for (c) blue, (d) red and (e) vegetation bands show variations in the SNR for smoke detection in relation to the $\lambda$ of light being measured.

### 3.1.1 From Full Dataset $\mathcal{D}$ to Mie-Derived Dataset $\mathcal{D}_M$

Let $\mathcal{D} = \{\mathcal{X}, \mathcal{Y}\}$ be the original dataset, where each label $y_i \in \mathcal{Y}$ corresponds to multiple satellite images $[x_{(i,t_0)}, ..., x_{(i,t_N)}] \in \mathcal{X}$ over a given time window. Using Mie scattering principles, we select the image $x_{(i,t_M)}$ with the highest expected smoke SNR to form a one-to-one dataset $\mathcal{D}_M = \{\mathcal{X}_M, \mathcal{Y}\}$ such that $\mathcal{X}_M \subset \mathcal{X}$ and $|\mathcal{X}_M| = |\mathcal{Y}|$. Based on forward scattering criteria, the trivial strategy would be to pull imagery from GOES-West right after sunrise and from GOES-East right before sunset when the SZA is closest to $90°$. To avoid image artifacts caused by extreme SZA, we exclude scenes with SZA$> 88°$ [24]. The resulting dataset $\mathcal{D}_M$ (Table 3) contains over 200,000 samples where the satellite image is chosen based on which frame within the annotation time window would exhibit the strongest forward scattering geometry and thus the highest potential smoke SNR if smoke were present.

### 3.1.2 PLDR Dataset $\mathcal{D}_p$

The $\mathcal{D}_M$ data selection process introduces a potential bias for resulting models to limit smoke identification to higher SZAs. Additionally, $\mathcal{D}_M$ is limited to providing the timestamp for maximum possible smoke SNR, it does not give information to point to which image aligns best with the smoke label. To address these limitations, we propose using $\mathcal{D}_M$ as a intermediary dataset in the PLDR workflow (Figure 5) that will predict the satellite image that best matches the analyst's smoke annotation to produce $\mathcal{D}_p$.

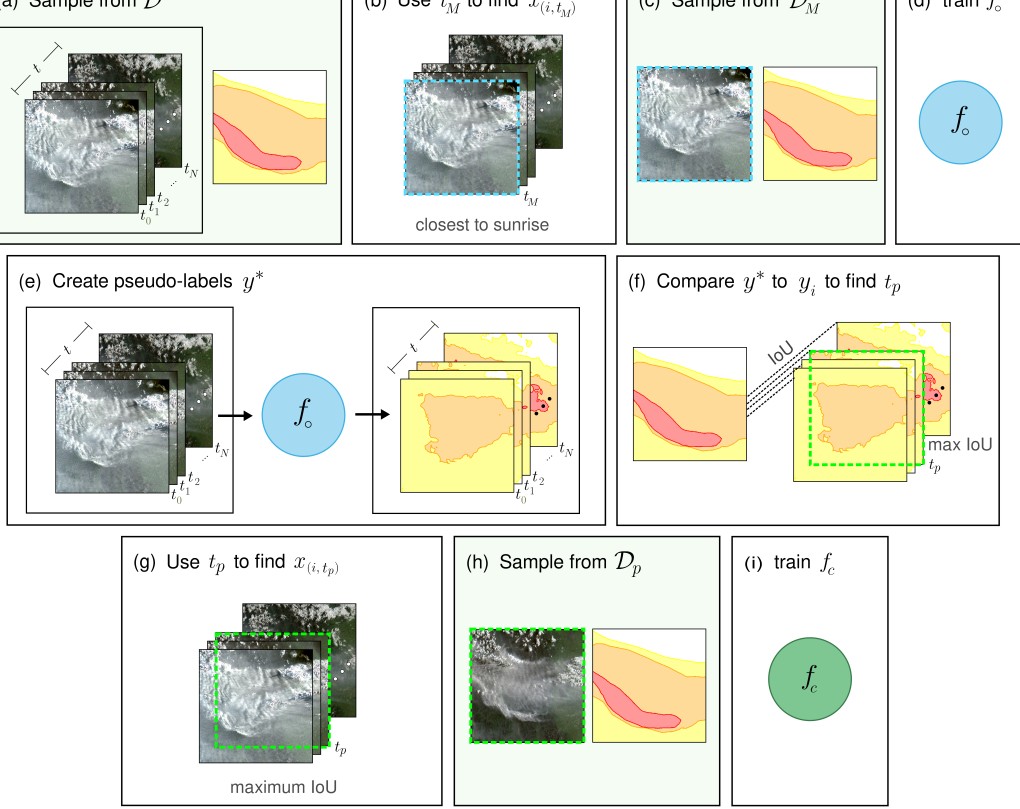

Figure 5: PLDR applied to create the SmokeViz dataset. Green boxes indicate dataset stages. (a) For original dataset $\mathcal{D}$ - analyst annotation $y_i$ corresponds to $N$ satellite images across time window $t$ so that $([x_{(i,t_0)}, ..., x_{(i,t_N)}], y_i) \in \mathcal{D}$; (b) use Mie scattering to find the time, $t_M$, that corresponds with satellite image $x_{(i,t_M)}$ that would produce the highest possible SNR if smoke was present; (c) resulting $\mathcal{D}_M$ is one-to-one $(x_{(i,t_M)}, y_i) \in \mathcal{D}_M$; (d) parent model $f_\circ$ is trained on $\mathcal{D}_M$ such that $f_\circ(x_{(i,t_M)}) = y_i$; (e) apply a greedy algorithm $f_\circ([x_{(i,t_0)}, ..., x_{(i,t_N)}]) = [y^*_{(i,t_0)}, ..., y^*_{(i,t_N)}]$ to create IPLs $y^*$ for each candidate image; (f) compute the intersection over union (IoU) between $y^*$ and $y_i$ to identify the time $t_p$ where the IPL and analyst annotation have the maximum IoU; (g) match $t_p$ to its corresponding image $x_{(i,t_p)}$ that is predicted to best match the analyst annotation; (h) SmokeViz dataset $\mathcal{D}_p$ created; (i) child model $f_c$ is trained on $\mathcal{D}_p$ such that $f_c(x_{(i,t_p)}) = y_i$ is used to detect and classify the density of wildfire smoke plumes in GOES imagery.

Table 2: A comparison of how smoke density would be represented by one-hot encoding commonly used for categorical data to thermometer encoding often used for ordinal data.

| density | one-hot | thermometer |
|---|---|---|
| none | [0 0 0] | [0 0 0] |
| light | [0 0 1] | [0 0 1] |
| medium | [0 1 0] | [0 1 1] |
| heavy | [1 0 0] | [1 1 1] |

Table 3: Dataset split for $\mathcal{D}_M$ and $\mathcal{D}_p$, samples for 2024 go up to November 1st. We use an entire year of data for both validation and testing sets to capture year-long wildfire trends.

| dataset | $\mathcal{D}_M$ | $\mathcal{D}_p$ | years |
|---|---|---|---|
| training | 158,391 | 127,197 | 2018-21, 24 |
| validation | 20,056 | 17,793 | 2023 |
| testing | 21,542 | 18,489 | 2022 |

To build the parent model $f_\circ$, that will create the intermediary pseudo-labels (IPLs), we implement `Segmentation Models PyTorch` [25] with EfficientNetV2 [26] as the encoder and PSPNet [27] as the decoder. Input images are $256 \times 256 \times 3$ true-color snapshots; the output is a $256 \times 256 \times 3$ classification map predicting categorical smoke density. We use thermometer encoding (Table 2)

and apply binary cross-entropy loss across density levels. Thermometer encoding is chosen over one-hot encoding because it captures the ordinal structure of smoke density categories (none < light < medium < heavy). In thermometer encoding, each higher class includes all lower class activations (e.g., heavy = [1 1 1]), allowing the model to learn not just class distinctions, but the relative severity of smoke. After performing a confidence threshold analysis discussed in the Supplementary Materials Section H, we use a confidence threshold of IoU > 0.1 to exclude samples with negligible overlap.

Figures 6 and 7 give statistical information on SmokeViz as well as highlight the possible influence of agricultural burns on the dataset distribution and possible model performance. Figure 6 shows that sample counts in SmokeViz peak in March and April, corresponding to agricultural burning rather than wildfire activity. During these months, IoU performance is relatively low in comparison to the scores observed from May through September which align with peak wildfire activity. Figure 7 further supports this trend, showing that the southeastern (SE) quadrant, where agricultural burns are prevalent, contributes 55% of all samples but exhibits relatively low IoU performance. These patterns suggest that agricultural burns, which are typically smaller in spatial extent and less visually distinct than large wildfires, present a greater challenge for accurate detection and segmentation by the model.

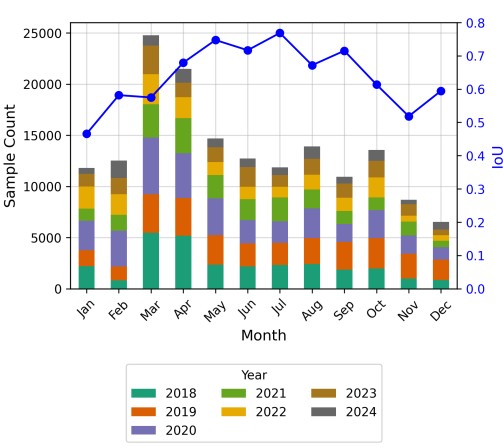

Figure 6: Monthly distribution of samples in the full dataset $\mathcal{D}_p$ (left), and monthly IoU scores between $f_c$ predictions and analyst annotations on the $\mathcal{D}_p$ test set (right).

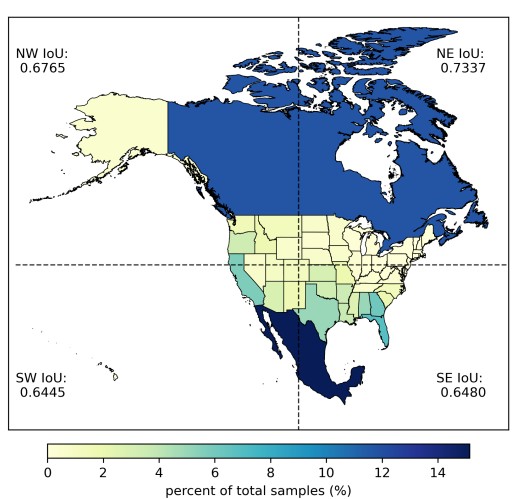

Figure 7: Sample percent contribution by region in the full $\mathcal{D}_p$ dataset and IoU performance of $f_c$ on the $\mathcal{D}_p$ test set across quadrants centered at $(40°\text{N}, -105°\text{W})$.

### 3.2 Benchmark Models

We benchmark the SmokeViz dataset $\mathcal{D}_p$ using PSPNet [27] with EfficientNetV2 [26], DPT [28] with ViT [29], Segfomer [30] and UperNet [31] with EfficientVit [32]. Each model is trained for 100 epochs while limited to 24 hours using a batch size of 256 and the Adam optimizer on 2 94GB Nvidia H100-NVL GPUs. These semantic segmentation architectures are selected for their relatively low memory requirements and effectiveness in segmenting multi-scale objects such as smoke plumes.

## 4 Results

We evaluate the performance of the parent ($f_o$) and child ($f_c$) models using Intersection over Union (IoU), precision and recall metrics on the test sets of both $\mathcal{D}_M$ and SmokeViz ($\mathcal{D}_p$), as shown in Table 4. For each smoke density class, IoU is calculated as the pixel-level intersection between model predictions and HMS analyst labels, divided by their union, aggregated over all test samples.

Table 4: Segmentation metrics comparing $f_\circ$ and $f_c$ on the $\mathcal{D}_M$ and $\mathcal{D}_p$ test sets.

| | $f_\circ$ | | $f_c$ | |
|---|---|---|---|---|
| Metric | $\mathcal{D}_M$ | $\mathcal{D}_p$ | $\mathcal{D}_M$ | $\mathcal{D}_p$ |
| Heavy IoU | 0.2600 | 0.2909 | 0.2685 | **0.3804** |
| Medium IoU | 0.3465 | 0.3765 | 0.3496 | **0.4776** |
| Light IoU | 0.5067 | 0.5675 | 0.5276 | **0.7301** |
| Overall IoU | 0.4574 | 0.5076 | 0.4831 | **0.6532** |
| Precision | 0.7452 | 0.8221 | 0.7319 | **0.8204** |
| Recall | 0.5422 | 0.5702 | 0.5869 | **0.7622** |

As shown in Table 4, in terms of IoU, $f_c$, that was trained on $\mathcal{D}_p$, consistently outperform $f_\circ$, that was trained on $\mathcal{D}_M$, across all smoke density categories. For both $f_\circ$ and $f_c$, IoU improves when evaluated on $\mathcal{D}_p$. The highest overall IoU = 0.6532, is achieved by $f_c$ on $\mathcal{D}_p$, indicating that PLDR improves image-label alignment and reduces training noise.

The increase in recall across both models, particularly for $f_c$ on $\mathcal{D}_p$, indicates that the PLDR-based dataset improves the model's ability to detect true smoke pixels. This suggests that training on the PLDR dataset enhances the model's sensitivity without a significant sacrifice in precision. In the context of smoke plume detection, favoring recall over precision can be desirable, as undetected smoke plumes can lead to errors in atmospheric and air-quality assessments.

Figure 8 illustrates a case in which the PLDR-selected frame better represents the HMS annotation than the Mie-derived selection. Here, the heavy smoke IoU improves from 0.01 to 0.59. While the Mie-derived image is selected based on its proximity to sunrise, PLDR chooses the frame with the highest overlap between the model-generated intermediary pseudo-label and the analyst annotation. This example highlights PLDR's advantage in resolving temporal ambiguity.

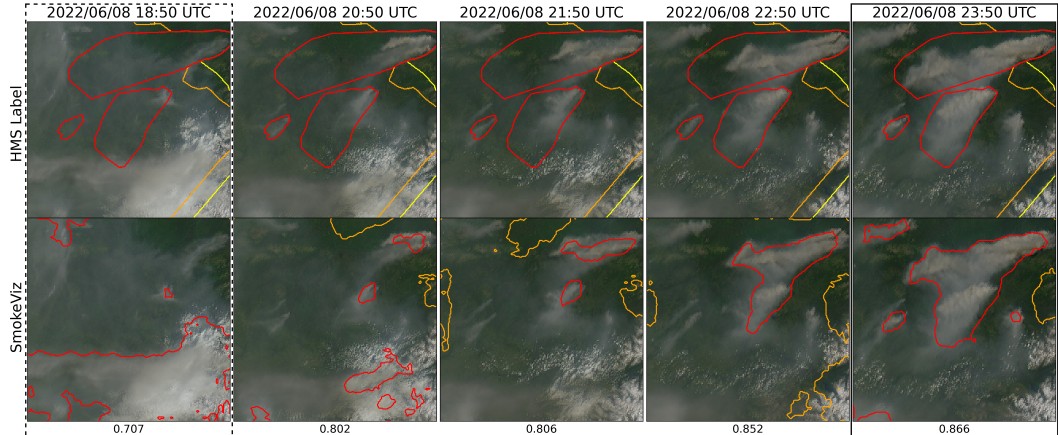

Figure 8: GOES-West imagery from June 8, 2022, over Alaska (61.06°N, 156.12°W). Daylight spanned 12:43-7:53 UTC. The single static HMS annotation (top row) spans 18:50-23:50 UTC is compared with $f_\circ$-generated per-frame smoke predictions (bottom row). The leftmost frame (dotted) represents the Mie-derived image; the rightmost frame (solid) was selected via PLDR and achieves higher IoU.

To further examine the performance of $f_c$, we can qualitatively compare its predictions against HMS annotations for samples from $\mathcal{D}_p$ in Figure 9. The model outputs capture more spatially detailed and coherent smoke boundaries compared to the coarser, polygon-based analyst labels.

Table 5: Comparison of segmentation benchmark model IoU metrics on the SmokeViz dataset. Note that the first column is $f_c$.

| encoder decoder | EfficientNet[26] PSPNet[27] | EfficientViT[32] Segformer[30] | [32] UPerNet [31] | ViT [29] DPT[28] |
|---|---|---|---|---|
| Heavy | **0.3804** | 0.2602 | 0.3156 | 0.3292 |
| Medium | **0.4776** | 0.4378 | 0.4602 | 0.4420 |
| Light | **0.7301** | 0.6857 | 0.6960 | 0.7115 |
| Overall | **0.6532** | 0.6051 | 0.6173 | 0.6309 |

To benchmark performance across segmentation architectures, we evaluate several encoder-decoder models trained on $\mathcal{D}_p$. Table 5 reports IoU scores by smoke density and overall. PSPNet yields the best performance per density and overall. Results across models are relatively consistent, highlighting the robustness of the SmokeViz dataset for training diverse architectures.

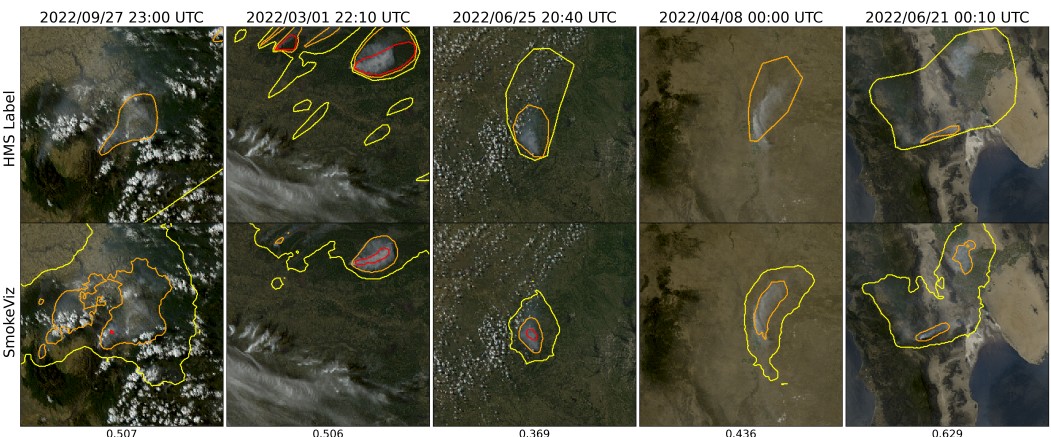

Figure 9: Examples of HMS annotations (top row) vs $f_c$ output (bottom row) on $\mathcal{D}_p$ samples. The overall IoU score is reported at the bottom of each column.

## 5 Limitations

Additional discussion and analysis of the primary limitations is provided in the Supplementary Materials. For traditional pseudo-labeling methods, the parent model may propagate biases into downstream models. In our case, as discussed in Section F, the increased detectability of forward-scattered light from smoke particulates may bias the model toward higher performance at larger solar zenith angles. Additionally, the HMS annotations do not distinguish between fire types and include a large number of controlled agricultural burns, which may limit the dataset's applicability for targeting large-scale wildfires, see Section D for more details.

Several additional limitations remain important directions for future work such as evaluating the model's ability to distinguish smoke from dust and investigating uncertainty in the analyst annotations.

## 6 Conclusion

In this study, we present **SmokeViz**, a refined satellite imagery dataset for semantic segmentation of wildfire smoke plumes. Starting from the original NOAA HMS annotations of coarse, many-to-one approximations of smoke boundaries, we transform the dataset into a one-to-one mapping between satellite frames and smoke annotations. While the Mie-derived dataset selection process maximized the potential for detecting smoke if present, it did not account for whether smoke was actually visible in the selected image, leading to a high incidence of label-image mismatch and associated training noise. To address this, we introduce **pseudo-label dimension reduction (PLDR)**, a physics-guided, semi-supervised method that uses a parent model trained on the Mie-derived dataset ($\mathcal{D}_M$) to generate

pseudo-labels across each annotation's time window. We then select the image with the highest spatial overlap between the intermediary pseudo-label and the HMS annotation to construct a refined dataset ($\mathcal{D}_p$). A child model trained on $\mathcal{D}_p$ achieves higher segmentation performance than the original parent model, as measured by IoU on both test sets, demonstrating the value of pseudo-label-based temporal alignment.

SmokeViz serves as a robust and representative dataset for training models to detect wildfire smoke in GOES imagery at the frame level. In addition to supporting real-time smoke segmentation, this dataset has potential applications in early wildfire detection, air quality monitoring, and as a smoke analysis product for data assimilation into dispersion models. It also provides a challenging benchmark for remote sensing models tasked with segmenting diffuse, low-contrast features like smoke. More generally, this work illustrates how PLDR can be used to resolve resolution mismatches between data and labels, especially in settings with time-series or video data paired with coarse annotations. The dataset is publicly available at `https://noaa-gsl-experimental-pds.s3.amazonaws.com/index.html#SmokeViz/` with code available at `https://github.com/reykoki/SmokeViz`.

## 7 Acknowledgments and Disclosure of Funding

This research was supported in part by NOAA cooperative agreement NA22OAR4320151, for the Cooperative Institute for Earth System Research and Data Science (CIESRDS). We thank Wilfrid Schroeder and the Hazard Mapping Systems team for giving guidance on how they created their smoke plume dataset. The statements, findings, conclusions, and recommendations are those of the author(s) and do not necessarily reflect the views of NOAA or the U.S. Department of Commerce.

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

## A   Original Data and Software Licenses

The HMS smoke product does not have a license attached to it. For GOES imagery, NOAA states: *"There are no restrictions on the use of this data".* PyTroll is distributed under the GNU General Public License v3.0, and Segmentation Models PyTorch is distributed under the MIT License.

## B   Satellite and Band Selection

To evaluate the impact of viewing geometry we show imagery from both GOES-East and GOES-West satellites (Figure 10). In this example, GOES-West provides great plume visibility near sunrise, consistent with Mie scattering physics, where forward-scattered light enhances aerosol contrast under favorable solar geometry.

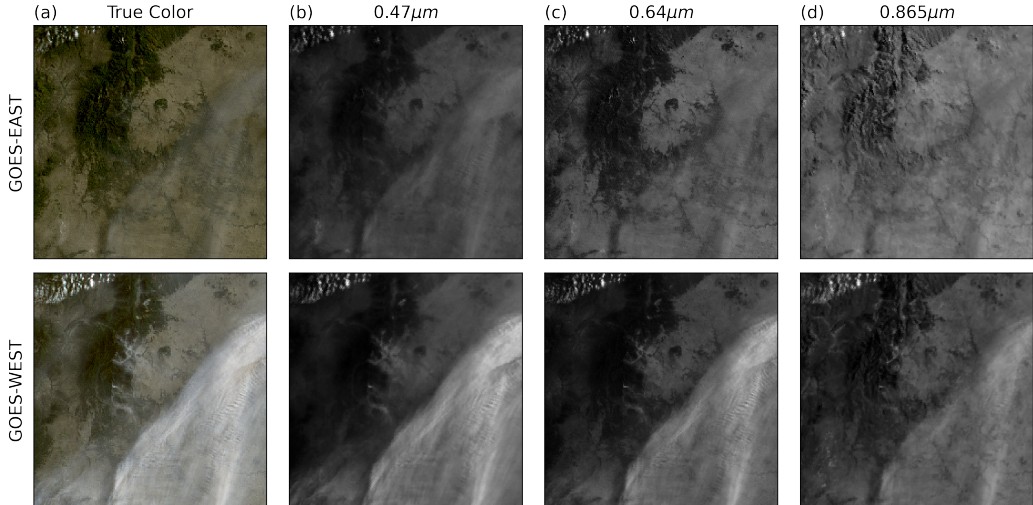

Figure 10: (a) True color GOES-EAST (top) and GOES-WEST (bottom) imagery from May $18^{th}$, 2022 centered at $(35.6°, -105.0°)$ in New Mexico, USA taken at 12:50 UTC. The GOES-East and West raw band imagery for (c) blue, (d) red and (e) veggie bands show variations in the SNR for smoke detection in relation to the $\lambda$ of light being measured.

As described in the main paper, Advanced Baseline Imager (ABI) bands 1-3 were selected for their high signal-to-noise ratio (SNR) and relevance to visible smoke. Figure 11 presents a smoke plume in a cloudy scene across all 16 ABI bands described in Table 6. The smoke signal is prominent in Bands 1–3 but diminishes in subsequent NIR channels. Band C07 (3.9 $\mu$m), which is sensitive to thermal anomalies, shows a strong fire signal at the source of the plume. While useful for active fire detection, including C07 for smoke segmentation may bias models toward learning fire-smoke co-location, reducing generalization to detached or low opacity smoke plumes, especially those classified as light density that have traveled far from the source. This concern supports the decision to limit input channels in SmokeViz to those that reflect the analyst operational view while minimizing potential modeling shortcuts and dataset size. The SmokeViz dataset development code is designed to be easily adapted to incorporate any desired spectral bands and/or composites.

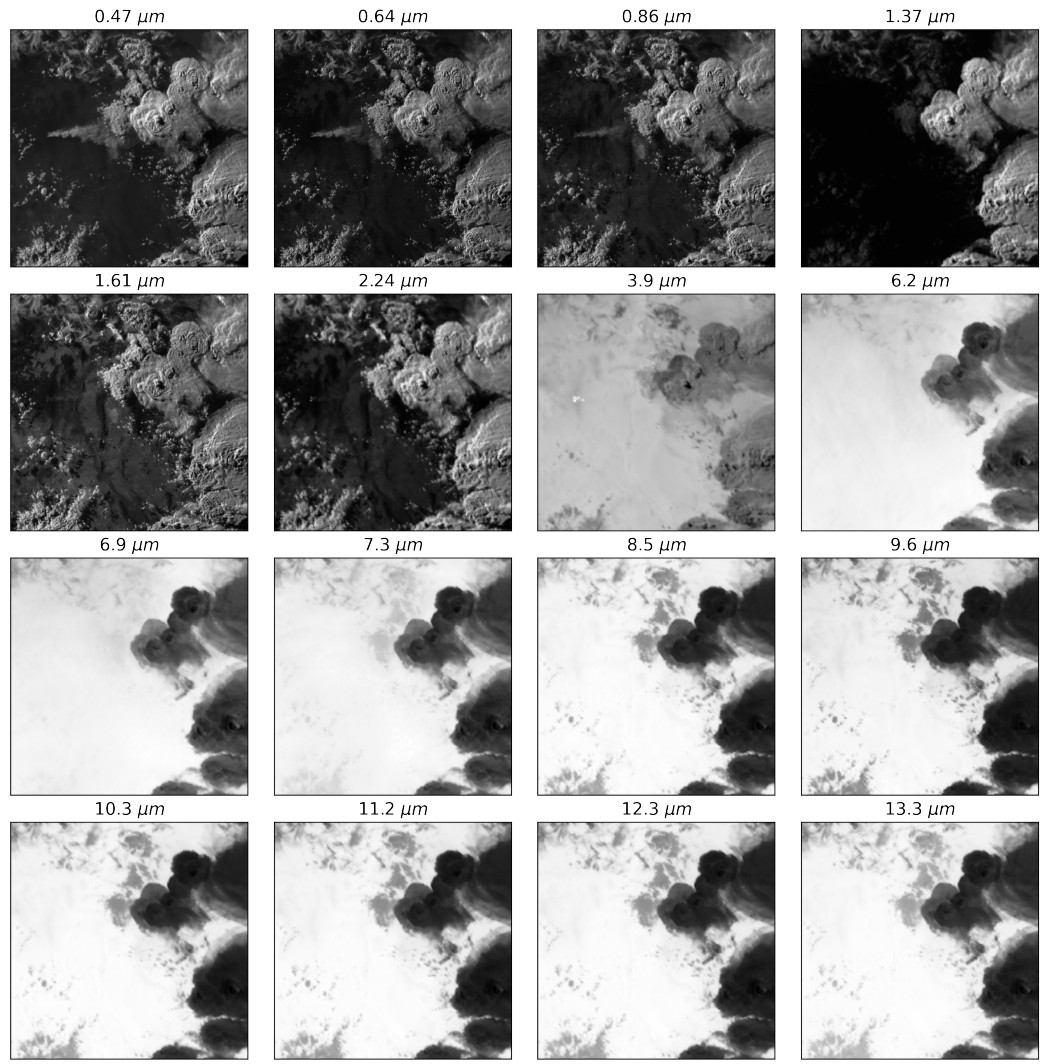

Figure 11: GOES-EAST imagery for all 16 bands from June $5^{th}$, 2022 centered at $(33.0°, -106.0°)$ in New Mexico, USA taken at 00:56 UTC.

Table 6: The GOES-Series Advanced Baseline Imager (ABI) provides data at 16 channels that cover visible (C01-C02), near-IR (C03-C06) and IR (C07-C16) bands.

| Band | Description | Center Wavelength ($\mu$m) | Spatial Resolution (km) |
|---|---|---|---|
| C01 | Blue visible | 0.47 | 1 |
| C02 | Red visible | 0.64 | 0.5 |
| C03 | Veggie near IR | 0.865 | 1 |
| C04 | Cirrus | 1.378 | 2 |
| C05 | Snow/Ice | 1.61 | 1 |
| C06 | Cloud particle | 2.24 | 2 |
| C07 | Shortwave IR | 3.9 | 2 |
| C08 | Upper-level water vapor | 6.2 | 2 |
| C09 | Mid-level water vapor | 6.9 | 2 |
| C10 | Lower-level water vapor | 7.3 | 2 |
| C11 | IR cloud phase | 8.5 | 2 |
| C12 | Ozone | 9.6 | 2 |
| C13 | Clean longwave IR | 10.35 | 2 |
| C14 | Longwave IR | 11.2 | 2 |
| C15 | Dirty longwave IR | 12.3 | 2 |
| C16 | $CO_2$ | 13.3 | 2 |

## C  Statistical Dataset Visualizations

Figures 12-14 summarize key statistical characteristics of the SmokeViz dataset.

Figure 12 presents a histogram of the number of GOES satellite frames associated with each HMS annotation. Since frames are available every 10 minutes, this visualization reflects the variability in annotation time window duration. Most annotations span between 5 and 50 frames, corresponding to 50 minutes to just over 8 hours, underscoring the need for resolving temporal ambiguity during dataset refinement.

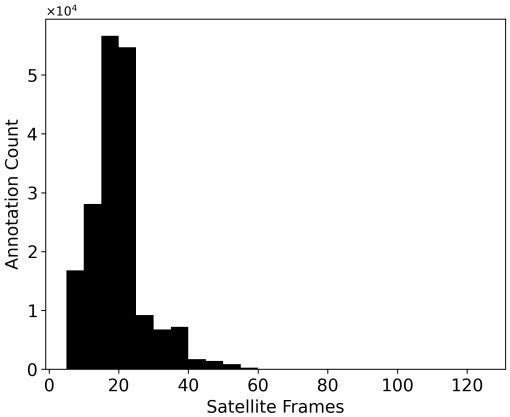

Figure 12: The number of annotations that span a number of satellite frames that are generated at a 10-minute interval.

Figure 13 shows the number of SmokeViz samples per year, stratified by smoke density. The year 2020 exhibits the highest sample count, aligning with an exceptionally active wildfire season across North America [33]. The density distribution across years also varies, with some years showing a higher relative proportion of light or medium smoke annotations.

Lastly, SmokeViz includes annotations across North America, Figure 14 summarizes the dataset's geographic coverage by country, including the United States, Canada, and Mexico.

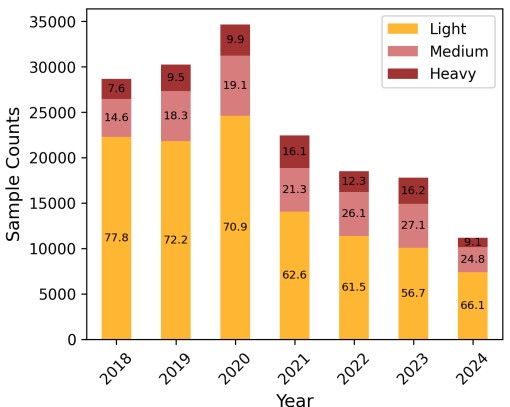

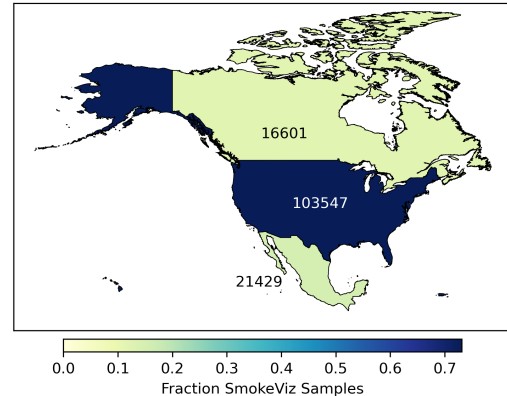

Figure 13: Annual sample counts in the Smoke-Viz dataset, broken down by smoke density class. Percentages within each column indicate the relative frequency of each density level for that year.

Figure 14: Geographic distribution of Smoke-Viz samples by center-pixel location: Canada (16,601), U.S. (103,547), Mexico (21,429), and others (24,886).

## D  Agricultural Burns

The monthly peak in sample counts, shown in Figure 6 and 15, occurs in March and April, preceding the typical wildfire season, which spans from late spring through fall. This early-season spike is likely due to prescribed agricultural burns, which are commonly conducted before vegetation exits winter dormancy [34]. Since HMS annotations do not distinguish between prescribed burns and wildfires, both event types are included in the dataset.

Model performance over time, shown in Figure 6 and 15, reveals that the highest IoU values for $f_c$ on the $\mathcal{D}_p$ test set occur during the peak wildfire season (May–September), not during the months with the highest sample counts. This suggests that prescribed burns that are typically smaller and less visually distinct (Light density) than large wildfires, are more difficult for the model to segment.

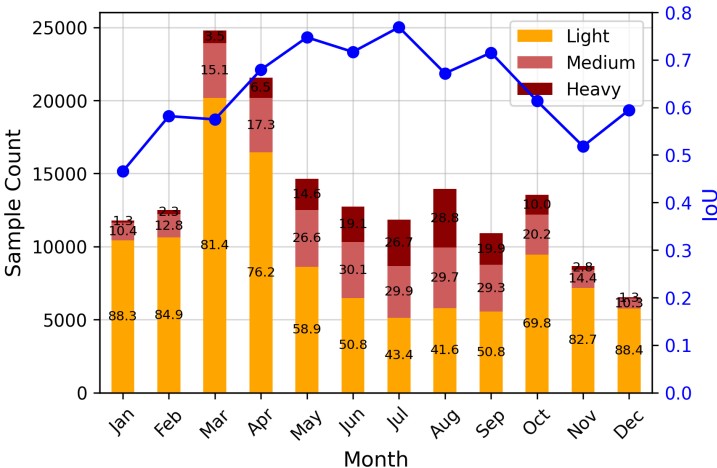

Figure 15: Monthly distribution of SmokeViz samples by smoke density class (Light, Medium, Heavy; bars; left axis) and corresponding overall IoU scores (blue line; right axis).

A spatial breakdown of sample density further supports this interpretation. Figure 7 in the main paper shows that the states with the highest number of samples are California, Georgia, and Florida. The elevated sample counts in southeastern states are consistent with regional practices of frequent prescribed burns. To investigate regional effects more explicitly, we divide the dataset into four

geographic quadrants: Northwest (NW), Southwest (SW), Northeast (NE), and Southeast (SE), centered around a continental midpoint at $(40°N, −105°W)$.

Table 7 reports test IoU and sample counts for each quadrant. Despite containing the largest share of training samples, the Southeast quadrant exhibits modest model performance. This degradation is likely due to the abundance of prescribed burns, which may produce smaller, low-opacity smoke plumes that are harder to detect. For SmokeViz users whose objective is to train models specifically for large wildfire detection, this highlights a limitation of the dataset: a substantial portion of the training data originates from controlled burns, which may not be representative of the intended task. One possible mitigation strategy is to filter out short-duration, light-density annotations (e.g., single-day events), though this is complicated by variability in analyst-defined time windows and labeling cadence per fire event.

Table 7: SmokeViz dataset sample distribution and $f_c$ test performance across geographic quadrants.

| Quadrant | SmokeViz Test Set IoU | SmokeViz Test Set Samples | SmokeViz Samples |
|---|---|---|---|
| Northwest (NW) | 0.6765 | 3,895 | 26,440 |
| Southwest (SW) | 0.6445 | 1,753 | 22,218 |
| Northeast (NE) | 0.7337 | 1,016 | 14,791 |
| Southeast (SE) | 0.6480 | 11,825 | 100,030 |

# E  Satellite Analysis

Figure 16 shows the distribution of samples from GOES-East and GOES-West in both the full SmokeViz dataset and the $\mathcal{D}_p$ test set, along with their respective segmentation performance using $f_c$. Although GOES-East contributes nearly three-quarters of all samples, model performance is substantially better on GOES-West test samples, with an IoU of 0.7270 compared to 0.6186 for GOES-East.

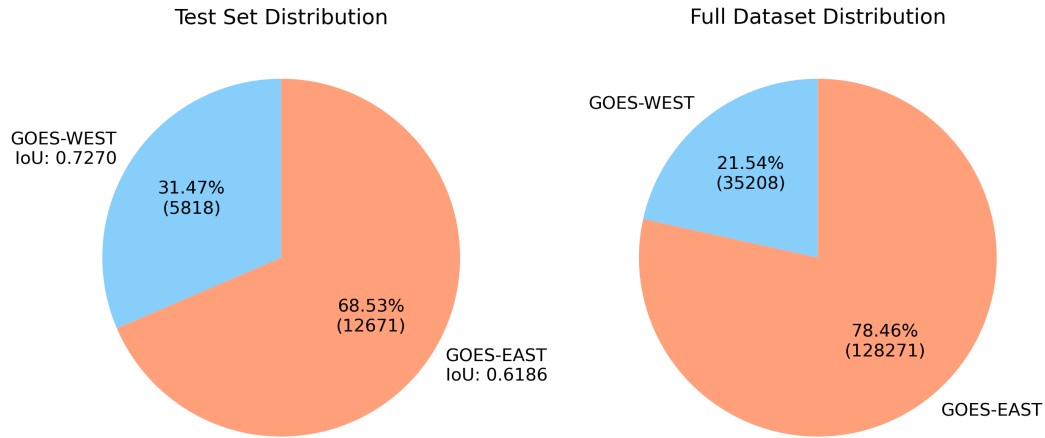

Figure 16: SmokeViz test set (left) and full dataset (right) sample distributions by satellite. GOES-West samples account for a smaller portion of the dataset but yield higher test IoU.

This discrepancy may stem from several factors. First, the observed signal quality varies between satellites depending on diurnal lighting, seasonal solar angles, and atmospheric conditions. GOES-West best captures forward-scattered sunlight during early morning hours over the western U.S., enhancing smoke visibility via Mie scattering and possibly boosting model accuracy. Additionally, sensor calibration, viewing geometry, and line-of-sight differences between the two platforms could contribute to systematic performance variation.

Another relevant factor is the operational transition between satellites. On June 18, 2022, GOES-17 was replaced by GOES-18 as the operational West satellite. While GOES-17 samples in the test set yield an IoU of 0.7380, GOES-18 samples yield a lower performance at 0.7003. This is likely due to the limited exposure of GOES-18 data during model training: training years (2018–2021) include only GOES-17, while GOES-18 is only present in the 2024 training data. This temporal imbalance may partially explain the drop in IoU for GOES-18.

Overall, these results suggest that while GOES-East offers broader coverage, the more favorable observational geometry of GOES-West, combined with consistent training data, produces stronger segmentation results. Future work may explore satellite-specific fine-tuning or normalization techniques to reduce these performance gaps.

## F   Sunset/Sunrise Bias

As discussed in the main paper's limitations, there exists a potential observational bias toward imagery captured near sunrise or sunset. This bias may originate from both our Mie-derived dataset ($\mathcal{D}_M$) and the original HMS annotations. Due to Mie scattering, the sun-satellite-smoke geometry results in a higher signal-to-noise ratio (SNR) when the solar zenith angle is near $90°$, typically around sunrise and sunset. This light configuration enhances the visual detectability of smoke plumes, guiding our initial selection method and potentially the analyst annotations.

In contrast, diurnal fire activity patterns . Wildfires tend to exhibit peak fire radiative power around solar noon due to increased temperature and wind [35], meaning that smoke intensity and spread are often greatest in midday imagery. The tension between observational clarity and fire behavior complicates dataset curation.

Figure 17 compares the performance of models trained on $\mathcal{D}_M$ (left) and the refined PLDR-generated dataset $\mathcal{D}_p$ (right), segmented by image with proximity to sunrise/sunset versus midday. The Mie-derived dataset favors high-SZA imagery, reflected in stronger IoU for morning/evening frames. In contrast, the SmokeViz dataset ($\mathcal{D}_p$), which selects the frame with the best overlap between model prediction and annotation, shows higher IoU for midday images, suggesting it more accurately aligns with fire dynamics rather than SNR optimization bias.

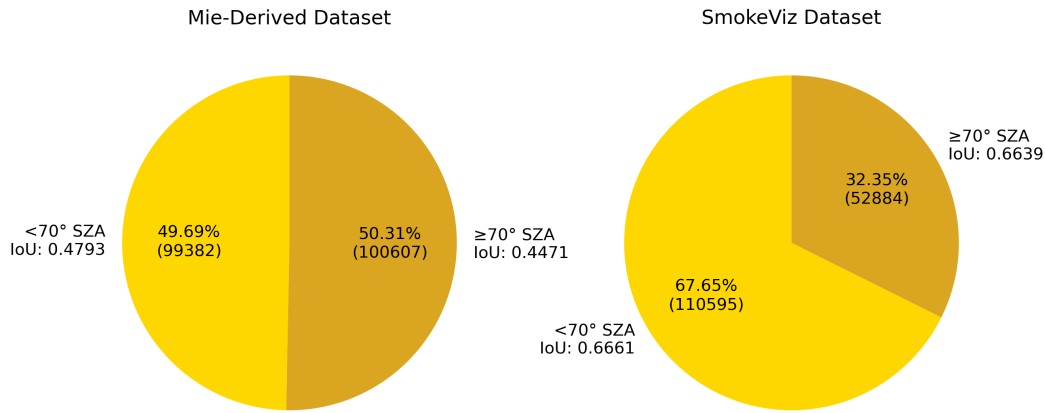

Figure 17: Distribution of data samples in the Mie-derived dataset ($\mathcal{D}_M$, left) and SmokeViz dataset ($\mathcal{D}_p$, right), split by if the solar zenith angle (SZA) is less than or greater than $70°$. $>70°$ SZA corresponds with the first/last $\approx 20\%$ of daylight, the remaining $\approx 60\%$ of midday daylight is represented in $<70°$ SZA. The PLDR refinement leads to large distribution and improved performance for midday samples.

This shift in temporal preference is further quantified in Figure 18, which shows the distribution of frame differences between corresponding samples in $\mathcal{D}_M$ and $\mathcal{D}_p$. Over 75% of the annotations were

assigned to different satellite frames between the two datasets. This illustrates the degree of temporal refinement enabled by PLDR, which prioritizes semantic alignment over SNR.

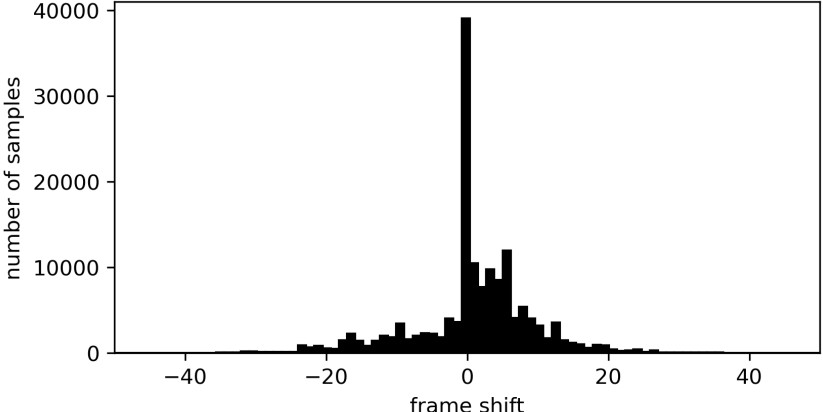

Figure 18: Histogram showing the shift in satellite frame selection between the Mie-derived dataset $\mathcal{D}_M$ and the PLDR-selected SmokeViz dataset $\mathcal{D}_p$. A value of 0 indicates the same frame was selected in both datasets. Approximately 25% of samples (39,044) share the same frame, reflecting substantial temporal reassignment between datasets.

## G  Qualitative Analysis on Performance

To complement the quantitative evaluation, we provide qualitative comparisons between the SmokeViz model predictions and the original HMS analyst annotations. These examples illustrate strengths and failure modes observed in the dataset.

Figure 19 presents five cases where $f_c$ does not perform optimally. In the first column, the model mistakenly identifies a smoke-like cloud as light-density smoke, a rare misclassification that reveals a potential weakness in cloud/smoke differentiation under certain lighting and texture conditions. In the remaining columns, SmokeViz underestimates the extent or density of visible smoke plumes. These errors generally occur in scenes with faint or fragmented plumes, where the signal-to-noise ratio is lower, or where overlapping atmospheric conditions make segmentation more difficult. In some cases, partial occlusion or lower contrast in the plume edges may have limited the model's confidence.

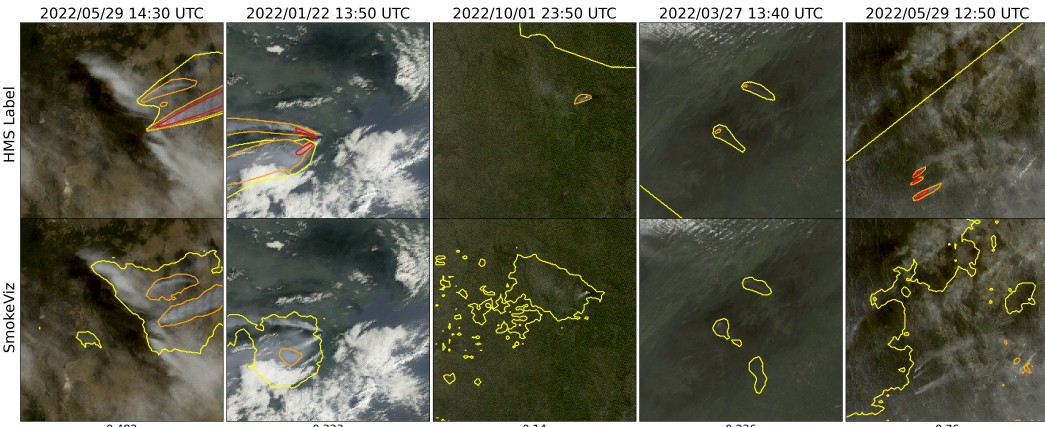

Figure 19: Challenging examples where SmokeViz underperforms. Top row: HMS annotations. Bottom row: SmokeViz model predictions. Leftmost column shows a false positive (cloud misclassified as smoke), remaining examples show under-segmentation or missed detection of smoke plumes. IoU values for each column are shown below.

In contrast, Figure 20 shows examples where SmokeViz closely matches the HMS labels or even outperforms them in delineating plume boundaries. In these scenes, the model produces tighter boundaries that conform well to the visible smoke extent, including fine structural details that the coarser analyst polygons often miss.

Together, these examples show that while the model performs robustly across many diverse smoke scenes, it still faces challenges in edge cases involving ambiguous cloud formations, light plumes, or visually occluded conditions. These qualitative insights can help inform aspects of improvements in future models.

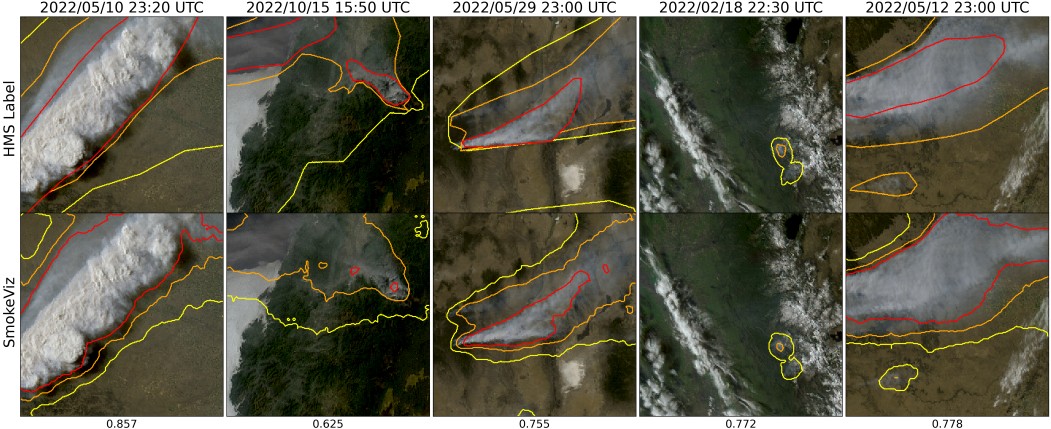

Figure 20: Examples of successful segmentation by SmokeViz. The model predictions (bottom) align well with HMS annotations (top), capturing plume shape and density more precisely. IoU scores shown below each sample indicate high overlap.

## H Confidence Threshold

To identify the optimal IoU threshold for constructing a PLDR-derived dataset, we trained separate segmentation models using datasets generated at a range of PLDR thresholds (y-axis) and evaluated each model across all test-set thresholds (x-axis), as shown in Figure 21. This full cross-evaluation isolates how well models trained on data of differing label quality generalize to both cleaner and noisier test conditions.

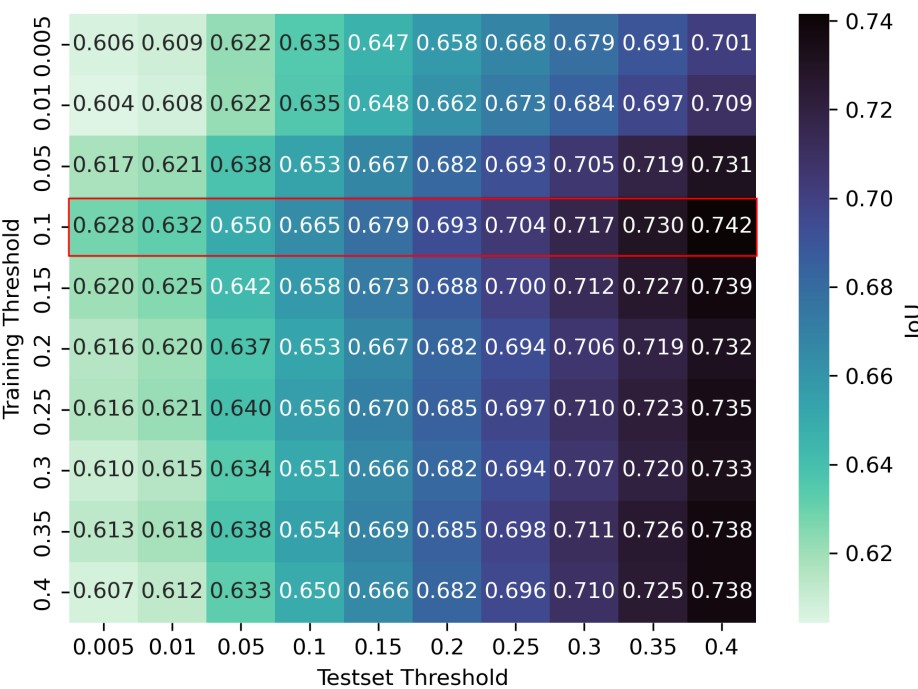

Figure 21: Overall IoU scores from models trained on training datasets constructed using PLDR at different IoU thresholds (y-axis) and evaluated across test sets generated with varying thresholds (x-axis).

Rather than maximizing performance only along the diagonal (where training and test thresholds match), this approach reveals which training threshold produces models that are robust to varying levels of label noise. The model trained with a threshold of 0.1 achieves consistently high IoU across a wide range of test thresholds, indicating strong generalization. Accordingly, an IoU threshold of 0.1 was selected for PLDR generation, as it yields models that maintain accuracy under both noisy and clean evaluation conditions, balancing label inclusiveness and reliability.

# I  Machine Learning Reproducibility

All relevant code for dataset construction, model training, and evaluation is publicly available at `https://github.com/reykoki/SmokeViz` to ensure transparency and reproducibility.

The models presented in this study are not optimized for state-of-the-art performance but are designed to serve two primary purposes: (1) generate pseudo-labels used in the PLDR workflow to construct the SmokeViz dataset, and (2) evaluate the relative performance of models trained on the Mie-derived dataset ($\mathcal{D}_M$) versus the refined SmokeViz dataset ($\mathcal{D}_p$).

To select the parent model $f_o$, we trained several encoder-decoder architectures on $\mathcal{D}_M$ and selected the one that achieved the highest overall Intersection over Union (IoU), as shown in Table 8. The top-performing model used EfficientNetV2 as the encoder and PSPNet as the decoder. This architecture was then used to generate intermediary pseudo-labels for the PLDR process.

Table 8: Comparison of segmentation model IoU metrics on the Mie-derived dataset ($\mathcal{D}_M$). The highest overall IoU model (EfficientNetV2 + PSPNet) was selected as $f_\circ$.

| **encoder** **decoder** | EfficientNet[26] PSPNet[27] | EfficientViT[32] Segformer[30] | [32] UperNet [31] | ViT [29] DPT[28] |
|---|---|---|---|---|
| Heavy | 0.2600 | 0.2266 | 0.1963 | 0.2220 |
| Medium | 0.3465 | 0.3067 | 0.3137 | 0.3647 |
| Light | 0.5067 | 0.4220 | 0.4406 | 0.4546 |
| Overall | 0.4574 | 0.3835 | 0.3970 | 0.4124 |

All models were trained distributed on 2 94GB H100s using the Adam optimizer with a learning rate of $1 \times 10^{-4}$, batch size of 256, and 100 training epochs. Hyperparameter values were chosen based on memory constraints and general suitability for large-scale semantic segmentation tasks.

### I.1 Datasheet for SmokeViz

Questions from the https://arxiv.org/abs/1803.09010 paper, v7.

#### I.1.1 Motivation

The questions in this section are primarily intended to encourage dataset creators to clearly articulate their reasons for creating the dataset and to promote transparencyabout funding interests.

**For what purpose was the dataset created?**

SmokeViz was created to serve as a large labeled dataset to be used in creating wildfire smoke plume related machine learning models. Applications include wildfire smoke detection or smoke dispersion modeling.

**Who created the dataset (e.g., which team, research group) and on behalf of which entity (e.g., company, institution, organization)?**

Collaborators are dispersed across multiple entities (see author list).

**Who funded the creation of the dataset?**

CIRES at University of Colorado Boulder and the National Oceanic and Atmospheric Administration Global Systems Laboratory funded this work.

**Any other comments?**

None.

#### I.1.2 Composition

Most of these questions are intended to provide dataset consumers with the information they need to make informed decisions about using the dataset for specific tasks. The answers to some of these questions reveal information about compliance with the EU's General Data Protection Regulation (GDPR) or comparable regulations in other jurisdictions.

**What do the instances that comprise the dataset represent (e.g., documents, photos, people, countries)?**

Each instance is a 256x256x3 RGB image from GOES imagery with an accompanying 256x256x3 binary masks corresponding to density of smoke. There are 3 densities of smoke - Light, Medium and Heavy.

**How many instances are there in total (of each type, if appropriate)?**

There are 163,479 samples - 111,621 for light, 33,570 for medium and 18,288 for heavy density smoke.

**Does the dataset contain all possible instances or is it a sample (not necessarily random) of instances from a larger set?**

The entire possible number of samples between 2018/01/01 - 2024/11/01 is 210,702. The dataset is reduced to 207,106 samples after filtering out any samples with no corresponding satellite imagery available or imagery that is less than 10 or over 90 percent saturation. Total saturation is defined when each pixel value is equal to 1. Mentioned in more detail in the paper, the dataset was further reduced down to 163,479 samples after applying a 0.1 IoU threshold during the PLDR process.

**What data does each instance consist of?**

The data is processed to correct for Rayleigh scattering, solar zenith angle and projected so each pixel is representative of the same area of land. The algorithm is referenced in the SmokeViz paper.

**Is there a label or target associated with each instance?**

Yes, there are no samples that are intended to not display any smoke.

**Is any information missing from individual instances?**

We have seen imagery where smoke is labeled but there's adjacent smoke plumes that were unlabeled. With human labels comes human errors.

**Are relationships between individual instances made explicit (e.g., users' movie ratings, social network links)?**

Some instances can overlap in geographic location, there can be multiple smoke plumes in one instance, but the index of the HMS smoke annotation is listed and can be mapped back to the original dataset for geolocational information.

**Are there recommended data splits (e.g., training, development/validation, testing)?**

We recommend using full years of data for training, validation and testing to keep full year long patterns of wildfire behavior. We use 2018-2021 and 2024 for training, 2023 for validation and 2022 for testing.

**Are there any errors, sources of noise, or redundancies in the dataset?**

The HMS smoke annotations that are used as truth are a source of noise as explained in the SmokeViz paper. These include approximations of smoke polygons mismatching actual location and time windows being too large that smoke moves during the time window. There is also noise caused by atmospheric interactions with light. Redundancies occur when there more than one smoke plume and annotation in one image.

**Is the dataset self-contained, or does it link to or otherwise rely on external resources (e.g., websites, tweets, other datasets)?**

The dataset is self-contained.

**Does the dataset contain data that might be considered confidential (e.g., data that is protected by legal privilege or by doctor-patient confidentiality, data that includes the content of individuals' non-public communications)?**

No.

**Does the dataset contain data that, if viewed directly, might be offensive, insulting, threatening, or might otherwise cause anxiety?**

No.

**Does the dataset relate to people?**

No, not directly, although wildfires do affect people, these images are at 1km resolution and do not show enough detail to relate to people or infrastructure.

**Does the dataset identify any subpopulations (e.g., by age, gender)?**

No.

**Is it possible to identify individuals (i.e., one or more natural persons), either directly or indirectly (i.e., in combination with other data) from the dataset?**

No.

**Does the dataset contain data that might be considered sensitive in any way (e.g., data that reveals racial or ethnic origins, sexual orientations, religious beliefs, political opinions or union memberships, or locations; financial or health data; biometric or genetic data; forms of government identification, such as social security numbers; criminal history)?**

No.

**Any other comments?**

No.

### I.1.3   Collection process

The answers to questions here may provide information that allow others to reconstruct the dataset without access to it.

**How was the data associated with each instance acquired?**

The labels from HMS smoke product are not validated or verified but is used for AirNow air quality assessments. The GOES imagery is collected by the ABI sensor and is corrected for any anomalies and also converted from photon count to radiance values.

**What mechanisms or procedures were used to collect the data (e.g., hardware apparatus or sensor, manual human curation, software program, software API)?**

Original low temporal resolution annotations were manual human analyst curated. To create the high temporal resolution annotations, we use pseudo-labeling discussed in detail within the SmokeViz paper.

**If the dataset is a sample from a larger set, what was the sampling strategy (e.g., deterministic, probabilistic with specific sampling probabilities)?**

The HMS smoke analysts are only looking for smoke during the daytime and do avoid annotations during heavy cloud cover.

**Who was involved in the data collection process (e.g., students, crowdworkers, contractors) and how were they compensated (e.g., how much were crowdworkers paid)?**

The NOAA NESDIS employed analysts are compensated as salaried federal employees.

**Over what timeframe was the data collected?**

2018-2024

**Were any ethical review processes conducted (e.g., by an institutional review board)?**

No.

### I.1.4 Preprocessing/cleaning/labeling

The questions in this section are intended to provide dataset consumers with the information they need to determine whether the "raw" data has been processed in ways that are compatible with their chosen tasks. For example, text that has been converted into a "bag-of-words" is not suitable for tasks involving word order.

**Was any preprocessing/cleaning/labeling of the data done (e.g., discretization or bucketing, tokenization, part-of-speech tagging, SIFT feature extraction, removal of instances, processing of missing values)?**

The data was processed according to the GOES True Color paper referenced in the SmokeViz paper methods section. This includes atmospheric, Rayleigh corrections and estimation of a Green band.

**Was the "raw" data saved in addition to the preprocessed/cleaned/labeled data (e.g., to support unanticipated future uses)?**

The raw data is available from the NOAA AWS webpage. `https://registry.opendata.aws/noaa-goes/` The HMS smoke annotations are available here: `https://www.ospo.noaa.gov/products/land/hms.html`

**Is the software used to preprocess/clean/label the instances available?**

Yes, Pytroll implements the algorithm discussed in the GOES True Color paper referenced in the SmokeViz paper.

**Any other comments?** None.

### I.1.5 Uses

These questions are intended to encourage dataset creators to reflect on the tasks for which the dataset should and should not be used. By explicitly highlighting these tasks, dataset creators can help dataset consumers to make informed decisions, thereby avoiding potential risks or harms.

**Has the dataset been used for any tasks already?**

It was used to train benchmark models mentioned in the paper that apply semantic segmentation to identify and classify smoke in satellite imagery.

**Is there a repository that links to any or all papers or systems that use the dataset?**

No.

**What (other) tasks could the dataset be used for?**

A machine learning based smoke dispersion forecast model, automated wildfire smoke detection and segementation, a smoke analysis product for data assimilation into smoke or air quality models.

**Is there anything about the composition of the dataset or the way it was collected and preprocessed/cleaned/labeled that might impact future uses?** No.

**Are there tasks for which the dataset should not be used?** No.

**Any other comments?** None

### I.1.6 Distribution

**Will the dataset be distributed to third parties outside of the entity (e.g., company, institution, organization) on behalf of which the dataset was created?**

No.

**How will the dataset will be distributed (e.g., tarball on website, API, GitHub)?**

`https://noaa-gsl-experimental-pds.s3.amazonaws.com/index.html#SmokeViz/`

**When will the dataset be distributed?**

It is currently available.

**Will the dataset be distributed under a copyright or other intellectual property (IP) license, and/or under applicable terms of use (ToU)?**

No.

**Have any third parties imposed IP-based or other restrictions on the data associated with the instances?**

No.

**Do any export controls or other regulatory restrictions apply to the dataset or to individual instances?**

No.

**Any other comments?**

None.

### I.1.7 Maintenance

These questions are intended to encourage dataset creators to plan for dataset maintenance and communicate this plan with dataset consumers.

**Who is supporting/hosting/maintaining the dataset?**

National Oceanic and Atmospheric Administration Global Systems Laboratory is hosting the dataset on Amazon Web Services.

**How can the owner/curator/manager of the dataset be contacted (e.g., email address)?**

`rey.koki@colorado.edu`

**Is there an erratum?**

No.

**Will the dataset be updated (e.g., to correct labeling errors, add new instances, delete instances)?**

Yes, only to add new instances.

**If the dataset relates to people, are there applicable limits on the retention of the data associated with the instances (e.g., were individuals in question told that their data would be retained for a fixed period of time and then deleted)?**

Not applicable.

**Will older versions of the dataset continue to be supported/hosted/maintained?**

No, it is too large to keep multple versions.

**If others want to extend/augment/build on/contribute to the dataset, is there a mechanism for them to do so?**

The code to extend/augment/build is publicly available `https://github.com/reykoki/SmokeViz`. We encourage anyone that would like to contribute to SmokeViz to reach out to `rey.koki@colorado.edu`. **Any other comments?**

None

