# OpenReview forum: "SmokeViz: A Large-Scale Satellite Dataset for Wildfire Smoke Detection and Segmentation"
_NeurIPS.cc/2025/Datasets_and_Benchmarks_Track — NeurIPS 2025 Datasets and Benchmarks Track poster_

### Official Review · Reviewer_NvAz · 2025-06-27

**Rating:** 5
**Confidence:** 4

**Summary:**

They introduce a dataset for learning to detect and classify smoke in remote sensing data.  Because the human annotations are not temporally or spatially precise, they introduce a method they call pseudo-label dimension reduction (PLDR) to refine the human annotations.  Since the human annotation covers multiple timestamps, they first choose the likely best timestamp for smoke observation based on the sun-satellite alignment.  They train an initial model on this data, and re-identify the best timestamp based on the agreement between the initial model’s output and the human annotation.  This new dataset is then used to train the final model.

Using their final dataset, they train and evaluate a variety of SOTA segmentation models and find that EfficientNet + PSPNet yields the best results.

**Additional Feedback:**

L69 “data assimilation” I did not understand what they are referring to here.
L74 “a” -> “an”
Figure 4 and other places: "veggie bands" should this be "vegetation"?

**Dataset Code Accessibility:**

Partly

**Dataset Code Comments:**

They did not provide any documentation with the code and dataset.

**Ethical Comments:**

The data is of insufficient resolution for any identifiable information to retained.

**Ethical Considerations:**

No, there are no or only very minor ethics concerns

**Final Justification:**

All of my concerns have been addressed adequately, so I raised my evaluation to "accept."

**Limitations Weaknesses:**

I would have liked to see more experimentation to validate the effectiveness of PLDR and other ideas like thermometer encoding in comparison to baselines.

I didn’t understand why the method was called pseudo-label dimension reduction as it doesn’t seem to have much connection to other dimension reduction algorithms such as PCA or an autoencoder.   They claim that PLDR is a general technique that could be used in other domains, but to me the method seemed specific to their use case since it requires some way of heuristically choosing the target image for the first step.

The paper was overall difficult to read, because of the large figures intruding on the text and the dense writing.  The paragraphs could be better organized into separate ideas rather than packing many different details into a single paragraph.

As far as I can tell the final step of PLDR, where they choose the target image based on IOU, was only described in the caption of Figure 5, not in the main text.  The caption of Figure 5 describes it as a “greedy algorithm” but I did not understand how this fit the pattern of a greedy algorithm.

The dataset and code are provided without any documentation, hindering re-use.

**Strengths Contributions:**

There are relatively few datasets for wildfire smoke detection in remote sensing data, and Table 1 convincingly shows the contribution of this new dataset, having two orders of magnitude more images than previous datasets.

The PLDR method addresses the issue of spatiotemporal precision in the human annotations with a reasonable and physics-informed approach.

They provide a thorough set of results, comparing both the initial and final models as well as different segmentation architectures, and present analyze failure cases as well as successes.

---

> ### Author Rebuttal · Authors · 2025-07-30
>
> ### General
> Thank you for your detailed and thoughtful review. We appreciate your recognition of the dataset’s scale and impact, the physical intuition behind PLDR, and the breadth of our model comparisons and analysis.
>
> We hope the following clarifications and outline for improvements in writing clarity and documentation strengthen your view of the work and support a recommendation above borderline accept.
>
> ### 3.1 Limitation
> I would have liked to see more experimentation to validate the effectiveness of PLDR and other ideas like thermometer encoding in comparison to baselines.
>
> ---
>
> ### Rebuttal 3.1
> We point the reviewer to L177-L181 and Table 2 for explanation and justification of using thermometer encoding over the more ubiquitous one-hot encoding. While we expect poor convergence for the reasons listed in the paper one why one-hot encoding is not suitable for this problem, we agree that it can solidify the claims to provide numerical results. We will add these results in the Supplementary Materials.
>
> One-hot vs thermometer encoding results on the Mie-Derived Dataset
>
>
> | Encoding Method        | Dataset       | Heavy IoU | Medium IoU | Low IoU | Overall IoU |
> |------------------------|---------------|-----------|------------|---------|--------------|
> | One-Hot Encoding       | Mie-derived    | 0.0012    | 0.0142     | 0.1470  | 0.1296       |
> | Thermometer Encoding   | Mie-derived    | 0.1510    | 0.2572     | 0.3933  | 0.3483       |
> | Difference             | –              | +0.1498   | +0.2430    | +0.2463 | +0.2187      |
>
>
>
> ---
>
> ### 3.2 Limitation
>
> I didn’t understand why the method was called pseudo-label dimension reduction as it doesn’t seem to have much connection to other dimension reduction algorithms such as PCA or an autoencoder. They claim that PLDR is a general technique that could be used in other domains, but to me the method seemed specific to their use case since it requires some way of heuristically choosing the target image for the first step.
>
> ### Rebuttal 3.2
> We acknowledge that the connection between PLDR and dimension reduction methods like PCA or autoencoders should be made clearer. PLDR reduces dimensionality along the time axis, selecting a single frame from a multi-frame time window that best aligns with the annotation. While PCA or autoencoders extract key latent features, PLDR selects the temporally optimal input to maximize semantic relevance.
> Although our use of Mie-scattering physics to initialize the parent model is domain-specific, the framework of PLDR is generalizable. Any application involving many-to-one data-label relationships (e.g., video frames mapped to a single annotation) can adopt PLDR by defining a domain-appropriate heuristic for the initial selection step. We will add clarifying text and examples of potential applications in the camera-ready main and supplementary materials.
>
> ---
>
> ### 3.3 Limitation
> The paper was overall difficult to read, because of the large figures intruding on the text and the dense writing. The paragraphs could be better organized into separate ideas rather than packing many different details into a single paragraph.
>
> ### Rebuttal 3.3
> We appreciate this feedback and recognize the importance of clarity and readability. For the camera-ready version, we will improve paragraph structure by breaking down longer, detail-heavy paragraphs into smaller sections that focus on separate ideas. We will also adjust figure placement and sizing to ensure they complement rather than disrupt the narrative flow.
>
> ---
>
> ### 3.4 Limitation
> As far as I can tell the final step of PLDR, where they choose the target image based on IOU, was only described in the caption of Figure 5, not in the main text. The caption of Figure 5 describes it as a “greedy algorithm” but I did not understand how this fit the pattern of a greedy algorithm.
>
> ### Rebuttal 3.4
> We appreciate this observation and agree that this is a critical step that should be made explicit in the main text. For the camera-ready version, we will revise the Methods section to explicitly describe the IoU-based frame selection step of PLDR, rather than leaving this detail only in the Figure 5 caption.
>
> PLDR’s selection process is greedy because it focuses on maximizing alignment for each annotation one time window at a time, rather than solving for a global optimum over all frames.
>
> ---
>
> ### 3.5 Limitation
> The dataset and code are provided without any documentation, hindering re-use.
>
> ### Rebuttal 3.5
> We appreciate the reviewer’s concern about documentation. We will include detailed documentation with the camera-ready release.
>
>
>
> ---
>
> ### Additional feedback
> L69 “data assimilation” I did not understand what they are referring to here. L74 “a” -> “an” Figure 4 and other places: "veggie bands" should this be "vegetation"?
> L69 “data assimilation” - Thank you for pointing out the domain-specific terminology. We will clarify this term to explain that we are referring to the integration of observational data (e.g., satellite-derived smoke masks) into numerical models such as HRRR-Smoke to improve their initial conditions and forecast accuracy.
>
> L74 “a” -> “an” - We will correct this grammatical error.
>
> Figure 4 and “veggie bands” terminology - We agree that “veggie” is an informal term for the 0.865 μm near-infrared band (C03). We will replace “veggie bands” with “vegetation band” throughout the paper for clarity.

---

> > ### Comment · Reviewer_NvAz · 2025-08-04
> > **Re: Rebuttal by Authors**
> >
> > Thanks to the authors for their careful consideration of my review and response to my questions.
> >
> > I believe all of my concerns have been addressed adequately.

---

### Official Review · Reviewer_xmzu · 2025-06-28

**Rating:** 4
**Confidence:** 4

**Summary:**

This paper introduces SmokeViz, a large-scale satellite dataset for wildfire smoke detection and segmentation. The dataset contains over 180,000 annotated samples derived from GOES satellite imagery and NOAA expert annotations. The authors propose pseudo-label dimension reduction (PLDR) to address temporal ambiguity in original annotations by using intermediary pseudo-labels to select the satellite image that best matches analyst annotations, creating one-to-one image-label pairs.

**Additional Feedback:**

No more comments.

**Dataset Code Accessibility:**

Yes

**Dataset Code Comments:**

Dataset is publicly hosted on AWS, the code is available on GitHub, and experimental settings are well-documented. Authors provide sufficient technical details for reproduction.

**Ethical Considerations:**

No, there are no or only very minor ethics concerns

**Final Justification:**

Thank you for the author's rebuttal. Regarding the smoke annotation part, we may have different understandings. The author has addressed most of my concerns, and I will maintain my score.

**Limitations Weaknesses:**

1. Fig. 6 and Fig. 7 reveal that 55% of samples come from agricultural burns, which show relatively poor IoU performance. This suggests the dataset may not be well-suited for large-scale wildfire detection applications.

2. No detailed analysis of PLDR component contributions, such as Mie scattering selection vs random selection, or impact of different confidence thresholds.

3. The paper does not adequately address distinguishing smoke from dust and clouds, which is crucial for practical applications.

4. Upon examination of the dataset samples, the annotation quality appears inconsistent, particularly at smoke plume boundaries and tail regions where smoke transitions from visible to dispersed. The coarse HMS annotations may not capture the subtle gradients and fuzzy boundaries that characterize real smoke plumes, potentially limiting the utility of the dataset for training high-precision segmentation models. I am not sure if this is due to the number of samples I examined, or if it is an issue with my understanding of these smoke characteristics.

**Strengths Contributions:**

1. Wildfire smoke monitoring is indeed a critical public health issue, and this dataset could support real-time air quality assessment and early warning systems with clear societal benefits.

2. The 180k+ samples significantly exceed existing datasets.

3. PLDR addresses the temporal ambiguity problem in HMS annotations.

---

> ### Author Rebuttal · Authors · 2025-07-30
>
> ### General
> Thank you for your detailed review and for recognizing the importance of SmokeViz for public health applications, as well as the scale of the dataset and the contribution of PLDR in addressing temporal ambiguity.
>
> We appreciate your thoughtful concerns regarding agricultural burn representation, component analysis within PLDR, cloud and dust discrimination, and annotation precision. In this rebuttal, we clarify these points and provide supporting evidence. We hope this rebuttal strengthens your confidence in the paper and encourages consideration above the borderline threshold.
>
> ### 2.1 Limitation
> Fig. 6 and Fig. 7 reveal that 55% of samples come from agricultural burns, which show relatively poor IoU performance. This suggests the dataset may not be well-suited for large-scale wildfire detection applications.
>
> ### Rebuttal 2.1
>
> We believe there is a misunderstanding. While we state that 55% of all samples originate from the SE quadrant (L189), where agricultural burns are common practice, available metadata does not allow differentiation between these two types of fires (L232).
>
> Although our findings suggest that agricultural burns are more difficult to segment (L40 Supp), they are operationally important as these events significantly impact regional air quality and PM_{2.5}​ exposure, posing health risks comparable to wildfire smoke. We see a future high impact application of the SmokeViz dataset to be used to train early wildfire detection models. Similar to controlled agricultural burns, early stage wildfires start smaller in spatial extent and are less visually distinct before they form into larger wildfires. Including agricultural burns provides variability in smoke plume morphology rather than overfitting to the large-scale high smoke density patterns primarily represented by wildfire smoke samples.
>
> Importantly, our results show in Figure 6 that IoU scores are highest during peak wildfire season (May-September), indicating that the model effectively captures wildfire-specific plume characteristics despite the presence of agricultural burns (L41 Supp).
>
> ---
>
> ### 2.2 Limitation
> No detailed analysis of PLDR component contributions, such as Mie scattering selection vs random selection, or impact of different confidence thresholds.
>
> ### Rebuttal 2.2
> For Mie scattering vs random selection, we address why random selection would not be an informed choice since many frames within the analyst time window contain no smoke (L141, Figure 2). We agree that a numerical justification would only strengthen that argument and will include these results in the Supplementary Materials. We did not perform a study on the impact of varying confidence thresholds in the initial submission due to computational constraints. These limitations have since been resolved with updated and expanded computational resources, and we have begun running the experiments. We look forward to including the results in the camera-ready version.
>
> | Metric         | Random Choice | Mie-Derived |
> |----------------|----------------|-------------|
> | High IoU       | 0.265          | 0.265       |
> | Medium IoU     | 0.307          | 0.345       |
> | Low IoU        | 0.436          | 0.495       |
> | Overall IoU    | 0.398          | 0.446       |
> **Table 1** compares results of models trained on the Mie-derived dataset and random choice dataset.
>
> | IoU Threshold | High IoU | Medium IoU | Low IoU | Overall IoU | Overall IoU on .01 threshold test set |
> |---------------|----------|------------|---------|--------------|-------------------|
> | 0.001         | 0.347    | 0.440      | 0.629   | 0.569        | 0.575             |
> | 0.005         | 0.342    | 0.444      | 0.646   | 0.583        | 0.586             |
> | 0.01          | 0.323    | 0.453      | 0.657   | 0.590        | 0.590             |
> | 0.015         | 0.346    | 0.457      | 0.664   | 0.595        | 0.593             |
> | 0.05          | 0.331    | 0.459      | 0.691   | 0.616        | 0.599             |
> | 0.075         | 0.361    | 0.467      | 0.704   | 0.628        | 0.603             |
> | 0.1           | 0.362    | 0.454      | 0.721   | 0.636        | 0.603             |
> | 0.2           | 0.349    | 0.475      | 0.767   | 0.671        | 0.608             |
> **Table 2** shows results of varying the IoU threshold, we continue experiments to introduce larger IoU thresholds.
>
> ### 2.3 Limitation
> The paper does not adequately address distinguishing smoke from dust and clouds, which is crucial for practical applications.
>
> ### Rebuttal 2.3
>
> While SmokeViz primarily focuses on refining wildfire smoke annotations, it inherently includes challenging cloud-interference scenarios. Quantifying this performance is an outstanding grand challenge due to the absence of reliable cloud mask data, particularly when smoke and cloud overlap.
>
> Dust discrimination is a key future direction but is similarly constrained by limited and unreliable data products. Notably, existing operational models such as HRRR-Smoke [7] also do not perform these distinctions likely due to the lack of reliable data products.
>
> ### 2.4 Limitation
> Upon examination of the dataset samples, the annotation quality appears inconsistent, particularly at smoke plume boundaries and tail regions where smoke transitions from visible to dispersed. The coarse HMS annotations may not capture the subtle gradients and fuzzy boundaries that characterize real smoke plumes, potentially limiting the utility of the dataset for training high-precision segmentation models. I am not sure if this is due to the number of samples I examined, or if it is an issue with my understanding of these smoke characteristics.
>
> ### Rebuttal 2.4
>
> We appreciate this observation and believe it highlights an important characteristic of smoke data rather than a flaw in the dataset. As noted in L103, the diffuse boundaries of smoke plumes are inherently ambiguous, especially in tail regions where smoke transitions into thin haze. What may appear as inconsistent labeling may be more of a reflection of the intrinsic diffuse and transitional nature of smoke plumes rather than annotation errors. HMS annotations are produced by trained NOAA analysts whose expertise is trusted on a national level for operational wildfire and air quality monitoring. By incorporating the challenge of dispersed edges, SmokeViz provides a rare and valuable benchmark for the machine learning community, compared to previous benchmarks that segment only sharply bounded objects (L101) [16-20].
>
> We fear a misunderstanding, Table 4 in the main text shows our benchmark model obtaining high-precision performance (0.7907), indicating that SmokeViz is capable of training high-precision segmentation models.

---

> > ### Comment · Reviewer_xmzu · 2025-08-01
> >
> > Thank you for the author's rebuttal. Regarding the smoke annotation part, we may have different understandings. The author has addressed most of my concerns, and I will maintain my score.

---

### Official Review · Reviewer_doD8 · 2025-07-07

**Rating:** 4
**Confidence:** 4

**Summary:**

Paper Summary:

SmokeViz introduces the largest publicly available satellite dataset (183,672 samples) for wildfire smoke segmentation, derived from NOAA HMS annotations and GOES-East/West imagery. To resolve temporal mismatches in coarse HMS labels (spanning multi-hour windows), the authors propose Pseudo-Label Dimension Reduction (PLDR). This semi-supervised method trains a parent model on Mie scattering-optimized imagery, generates intermediary pseudo-labels (IPLs) across all candidate frames, and selects the best-aligning image via IoU maximization to create Dp. A child model trained on Dp achieves SOTA segmentation (IoU 0.525), enabling high-resolution smoke monitoring.

**Dataset Code Accessibility:**

Yes

**Dataset Code Comments:**

The authors have provided urls to access the codes and datasets.

**Ethical Considerations:**

No, there are no or only very minor ethics concerns

**Limitations Weaknesses:**

1. The parent model in PLDR relies on mie-scattering-selected images, which may propagate a 'high Solar Zenith Angle (SZA) preference' to the child model, reducing its generalization capability under non-ideal lighting conditions (e.g., midday). Additionally, as claimed in the limitations, the pseudo-label generation process does not evaluate the parent model's error propagation risk, potentially amplifying initial selection bias.

2. Half of samples originate from agricultural burns, whose smoke morphology differs significantly from wildfires (smaller area, lower concentration), potentially leading to poorer model performance in wildfire scenarios.

3. HMS analyst annotations exhibit subjective variations, yet inter-annotator consistency (e.g., Kappa coefficient) remains unquantified, potentially compromising label reliability assessment.

4. PLDR selects single frame image matching annotations through IPLs, but the HMS original annotation window is longer than 20 hours and the annotation is only an "approximate spatiotemporal range" of smoke. Although PLDR improves alignment quality, it does not analyze residual time misalignment errors or subjective labeling.

**Strengths Contributions:**

1. PLDR uniquely employ the pseudo-labeling strategy to resolve spatiotemporal annotation ambiguities—converting coarse, variable-length labels into precise frame-level pairs without new human annotation. This enables temporal alignment previously unaddressed in remote sensing .

2. SmokeViz is the largest smoke segmentation dataset, covering diverse North American biogeographies and fire seasons, enhancing model generalizability .

---

> ### Author Rebuttal · Authors · 2025-07-30
>
> ### General:
>
> Thank you for your thoughtful review and for highlighting the novelty of PLDR and the scale and diversity of the SmokeViz dataset. We appreciate your recognition of the contributions to temporal alignment and generalizability in smoke segmentation.
>
> You raised important concerns around SZA bias, agricultural burns, annotation subjectivity, and temporal misalignment. We hope by providing clarification and pointing you to supporting analysis strengthens confidence in the work and encourages you to consider increasing your score.
>
> ---
>
> ### 1.1 Limitation
> The parent model in PLDR relies on mie-scattering-selected images, which may propagate a 'high Solar Zenith Angle (SZA) preference' to the child model, reducing its generalization capability under non-ideal lighting conditions (e.g., midday). Additionally, as claimed in the limitations, the pseudo-label generation process does not evaluate the parent model's error propagation risk, potentially amplifying initial selection bias.
>
> ### Rebuttal 1.1
> Section F of the Supplementary Materials provides an analysis of the potential bias towards high SZA imagery. We describe the contrast between larger SZAs near sunrise/sunset, which yield higher smoke signal-to-noise ratios (SNR), and smaller SZAs around midday, which coincide with peak fire activity (L88 Supp). Although $\mathcal{D}_M$​ is initially constructed by selecting the image with the highest SNR, approximately 80% of samples are reassigned to different timestamps during the PLDR process (L98 Supp, Figure 17), which prioritizes alignment with analyst annotations.
>
> To evaluate generalizability to midday conditions (defined here as >2 hours from sunrise/sunset), Figure 16 shows that after applying PLDR, IoU on the SmokeViz test set is higher for midday samples (+0.03) than for sunrise/sunset samples. In contrast, the initial Mie-derived dataset shows a reduction in IoU (-0.12) for midday conditions. This improvement indicates that PLDR effectively corrects temporal misalignment and provides better performance, rather than amplifying the parent model's high-SZA bias.
>
> Regarding error propagation, there may be a misunderstanding. Traditional pseudo-labeling encounters error propagation from parent to child model when the parent model makes incorrect predictions when generating pseudo-labels and those incorrect labels are used to train the child model. The child model may then “learn” those mistakes and possibly amplify those errors rather than correcting them. PLDR differs from traditional pseudo-labeling; we do not use the parent model’s predictions as ground truth for training the child model, but instead use the original HMS analyst labels to train each model. This mitigates error propagation from parent to child model in comparison to traditional pseudo-labeling since the child model never touches the intermediary pseudo-labels. Error between predictions and ground truth is calculated against the original analyst labels, not against parent predictions and the child model’s intermediary pseudo-labels.
>
> If the reviewer’s concern on error propagation relates to bias in satellite frame selection, the SZA bias analysis in Section F of the Supplementary Materials addresses this risk.
>
> ---
>
> ### 1.2 Limitation
> Half of samples originate from agricultural burns, whose smoke morphology differs significantly from wildfires (smaller area, lower concentration), potentially leading to poorer model performance in wildfire scenarios.
>
> ### Rebuttal 1.2
> We believe there is a misunderstanding. While we state that 55% of all samples originate from the SE quadrant (L189), where agricultural burns are common practice, available metadata does not allow differentiation between these two types of fires (L232).
>
> Although our findings suggest that agricultural burns are more difficult to segment (L40 Supp), they are operationally important as these events significantly impact regional air quality and PM$_{2.5}​$ exposure, posing health risks comparable to wildfire smoke. We see a future high impact application of the SmokeViz dataset to be used to train early wildfire detection models. Similar to controlled agricultural burns, early stage wildfires start smaller in spatial extent and are less visually distinct before they form into larger wildfires. Including agricultural burns provides variability in smoke plume morphology rather than overfitting to the large-scale high smoke density patterns primarily represented by wildfire smoke samples.
>
> Importantly, our results show in Figure 6 that IoU scores are highest during peak wildfire season (May-September), indicating that the model effectively captures wildfire-specific plume characteristics despite the presence of agricultural burns (L41 Supp).
>
> ---
>
> ### 1.3 Limitation
> HMS analyst annotations exhibit subjective variations, yet inter-annotator consistency (e.g., Kappa coefficient) remains unquantified, potentially compromising label reliability assessment.
>
>
> ### Rebuttal 1.3
> While HMS annotations involve subjectivity, as do most annotations in computer vision tasks, they are created by trained and paid NOAA analysts who follow established protocols. These annotations are trusted at a national operational level for air quality and wildfire monitoring, which underscores their reliability and accuracy. In comparison to smaller datasets, the large scale of SmokeViz (180k+ samples) can mitigate the effect of annotator inconsistencies by enabling models to learn the statistical structure of smoke features. Additionally, the strong performance of $f_{c}$ across temporal (Figure 6) and spatial (Figure 7) spans suggest that the annotations are of sufficient quality. Future work in collaboration with HMS analysts could explore quantifying inter-analyst consistency, but we maintain that expert operational annotations provide the strongest ground truth currently available. A detailed study of annotation practices and consistency would be better suited for a venue like CSCW, similar to the approach taken by [Anjum, CSCW 2021] for multi-object tracking.
>
> ---
>
> ### 1.4 Limitation
> PLDR selects single frame image matching annotations through IPLs, but the HMS original annotation window is longer than 20 hours and the annotation is only an "approximate spatiotemporal range" of smoke. Although PLDR improves alignment quality, it does not analyze residual time misalignment errors or subjective labeling.
>
> ### Rebuttal 1.4
> We acknowledge that HMS annotations represent an approximate spatiotemporal range of smoke, with occasional outliers exceeding 20 hours. However, as shown in Figure 12 of the Supplementary Materials, the vast majority of annotations span much shorter periods, with an average window of 2.91 hours.
>
> The expertise of operational-level trained smoke plume analysts cannot be replicated through crowd-sourcing where annotators are unable to be trained at the same level to be able to identify subtle and complex smoke patterns.

---

> > ### Comment · Reviewer_doD8 · 2025-08-05
> >
> > I would like to thank the authors' response to my comments. Overall, my concerns are well addressed and thus I will keep the positive score.

---

### Author Response · Authors · 2025-08-05

Thank you to all reviewers for the thoughtful feedback and for engaging in the discussion. These comments have helped us in polishing a camera-ready version of the paper and we are glad that we were able to resolve all concerns.

---

### Note · Authors · 2025-08-15

We thank the reviewers for their thoughtful engagement and constructive feedback. Several important concerns were raised during the review process, which we have addressed in detail and incorporated into an improved version of the paper.

A concern raised by reviewers doD8 and xmzu involved the prevalence of agricultural burns in the dataset and their potential impact on generalization to wildfires. We clarified in our rebuttals that while 55% of samples are from the southeastern U.S., the metadata does not distinguish fire types. Including a range of plume morphologies, such as those from agricultural burns, may improve model performance on early-stage wildfires and support broader generalization. We also point to Fig. 6, Supp which shows the model achieves highest IoU during peak wildfire season, supporting the relevance of SmokeViz to wildfire specific events.

A second concern, again from doD8 and xmzu, involved the subjectivity and temporal imprecision of HMS annotations. We emphasized the operational expertise of NOAA analysts and noted that most annotations span a narrow window (~3 hours; Fig. 12, Supp). We clarified that PLDR addresses temporal ambiguity by selecting a single frame per annotation via greedy IoU maximization (now described in main text, not only in Fig. 5 caption). We also provided new analysis showing PLDR improves midday performance (Fig. 16, Supp) and mitigates Solar Zenith Angle bias from initial Mie-based selection.

Reviewers xmzu and NvAz requested additional ablation-style experiments to validate PLDR. In response, we added results comparing Mie-derived vs. random frame selection (Rebuttal 2.2), showing a clear benefit (+0.048 IoU). We also performed an IoU threshold sweep (Rebuttal 2.2) and demonstrated that thermometer encoding outperforms one-hot encoding by +0.22 IoU (Rebuttal 3.1), further justifying design choices in the pipeline.

Reviewer NvAz also raised concerns about writing clarity, terminology (e.g., “vegetation band” vs. “veggie band”), and documentation. We thank them for these suggestions and will revise figure placement and paragraph structure, and provide detailed code and dataset documentation in the camera-ready version.

We are encouraged that all reviewers indicated their concerns were resolved. We appreciate the opportunity to contribute this work to the NeurIPS community and are excited to submit a strengthened final version.

---

### Decision · Program_Chairs · 2025-09-18

**Decision:**

Accept (poster)

**Comment:**

All reviewers are in favor of acceptance. The proposed dataset is large, comprehensive and carefully annotated through an innovative use of PLDR that is not common (but should be, perhaps) in satellite imagery datasets. The wildfire problem is highly significant, with huge increases in fires and damage over the past few years and decades. The authors addressed reviewers' concerns in the rebuttal by adding ablation studies on the value of PLDR, and clarifying points about distinguishing between controlled vs. wild fires. The dataset and its underlying methodology should be a significant contribution.